# Interdependent iron and phosphorus availability controls photosynthesis through retrograde signaling

Hye-In Nam[1], Zaigham Shahzad[2], Yanniv Dorone[1,3], Sophie Clowez[1], Kangmei Zhao [1], Nadia Bouain[3], Katerina S. Lay-Pruitt [4,5], Huikyong Cho[3], Seung Y. Rhee [1✉] & Hatem Rouached [4,5,6✉]

Iron deficiency hampers photosynthesis and is associated with chlorosis. We recently showed that iron deficiency-induced chlorosis depends on phosphorus availability. How plants integrate these cues to control chlorophyll accumulation is unknown. Here, we show that iron limitation downregulates photosynthesis genes in a phosphorus-dependent manner. Using transcriptomics and genome-wide association analysis, we identify two genes, *PHT4;4* encoding a chloroplastic ascorbate transporter and *bZIP58*, encoding a nuclear transcription factor, which prevent the downregulation of photosynthesis genes leading to the stay-green phenotype under iron-phosphorus deficiency. Joint limitation of these nutrients induces ascorbate accumulation by activating expression of an ascorbate biosynthesis gene, *VTC4*, which requires bZIP58. Furthermore, we demonstrate that chloroplastic ascorbate transport prevents the downregulation of photosynthesis genes under iron-phosphorus combined deficiency through modulation of ROS homeostasis. Our study uncovers a ROS-mediated chloroplastic retrograde signaling pathway to adapt photosynthesis to nutrient availability.

[1] Department of Plant Biology, Carnegie Institution for Science, Stanford, CA, USA. [2] Department of Cell and Developmental Biology, John Innes Centre, Norwich NR4 7UH, UK. [3] Department of Biology, Stanford University, Stanford, CA 94305, USA. [4] Plant Resilience Institute, Michigan State University, East Lansing, MI 48824, USA. [5] Department of Plant, Soil, and Microbial Sciences, Michigan State University, East Lansing, MI 48824, USA. [6] BPMP, Univ Montpellier, CNRS, INRAE, Montpellier SupAgro, Montpellier, France. ✉email: srhee@carnegiescience.edu; rouached@msu.edu

Chloroplasts are sites of photosynthesis, whose function requires numerous proteins encoded in the nuclear genome[1]. Although plants tightly orchestrate chloroplast-to-nucleus signaling (retrograde control), it is poorly understood at the mechanistic level. In addition, the adequate accumulation of nutrients such as iron (Fe) in chloroplasts is required for their optimal performance[2,3]. Up to 80% of Fe in leaves is located in the chloroplasts[4,5], where its ability to donate and accept electrons plays a central role in electron transfer reactions[6]. Fe is found in all electron transfer complexes PSI, PSII, cytochrome b6f complex, and ferredoxins and is required for the biogenesis of cofactors such as hemes and iron–sulfur clusters[7,8]. Plants grown under Fe-deficient (−Fe) environments show chlorotic symptoms[9], and compromised photosynthesis[2,3]. However, chlorotic leaves can also develop under high-phosphorus (P) conditions, despite replete Fe levels[10], challenging the causal connection between Fe concentration and chlorophyll accumulation. Moreover, we recently reported that rice plants grown under a combined Fe and P deficiency (−Fe−P) do not exhibit a chlorotic phenotype[11]. These observations revealed a gap in our understanding of the interdependent effects of nutrient availability on regulating photosynthesis. Here, we addressed this issue through a combination of global gene expression analyses and genome-wide association studies (GWAS) to find expression quantitative trait loci (eQTLs) and uncovered a regulatory module that controls chlorophyll accumulation in response to Fe and P availability. This module involves an ascorbic acid (AsA) synthesis enzyme named VITAMINC4 (VTC4), a chloroplastic AsA transporter named PHOSPHATE TRANSPORTER 4;4 (PHT4;4), and a putative transcription factor named BASIC LEUCINE-ZIPPER 58 (bZIP58). The functioning of this module sheds light on the importance of chloroplast–nucleus communications under co-occurring nutrient deficiencies in controlling photosynthesis.

## Results

**Iron-induced chlorosis is independent of Fe status.** We previously reported that Fe-deficiency-induced chlorosis depends on P availability in rice[11]. To investigate whether the interdependent effects of Fe and P availability on chlorosis are conserved across monocot and eudicot species, we phenotyped *Arabidopsis thaliana* Col-0 (eudicot) and *Lemna gibba* (monocot), along with *Oryza sativa* (monocot), under different regimes of Fe and P availability. Fe deficiency (−Fe) caused chlorosis in all three species, but only in the presence of P (−Fe+P) (Fig. 1a–c). Quantification of chlorophyll content confirmed that −Fe significantly reduced the accumulation of chlorophyll in all three species (Fig. 1d). However, under −Fe−P conditions, chlorophyll content was comparable to control (+Fe+P) conditions in these species (Fig. 1d). Next, we focused on Arabidopsis to gain insights into the physiological and molecular processes underlying the stay-green phenotype (lack of chlorosis development) under −Fe−P conditions. First, we asked whether the stay-green phenotype under −Fe−P is caused by an increase of Fe levels in shoots. Plants grown in −Fe+P conditions decreased total Fe in shoots by twofold compared to +Fe+P conditions (Supplementary Fig. 1). On the other hand, under +Fe−P conditions, Fe levels increased by 2.2-fold relative to +Fe+P conditions (Supplementary Fig. 1). Surprisingly, Fe levels in plants grown under −Fe−P were reduced and indistinguishable from the Fe levels in −Fe+P conditions (Supplementary Fig. 1). Therefore, the stay-green phenotype under −Fe−P appears independent of shoot Fe status. Furthermore, examination of bioavailable $Fe^{2+}$ in leaves using the Turnbull/DAB (stains $Fe^{2+}$)[12,13] staining revealed enhanced accumulation of Fe under +Fe−P and reduced Fe under −Fe+P

compared to control (+Fe+P) (Fig. 1e). Under −Fe−P, Fe accumulation was also reduced compared to control and indistinguishable from −Fe+P, indicating that the higher chlorophyll content under −Fe−P compared to −Fe+P is not due to an increase in bioavailable Fe (Fig. 1e). Similarly, quantification using the phenanthroline method[14] detected no difference in bioavailable $Fe^{2+}$ accumulation under −Fe+P and −Fe−P conditions (Fig. 1f). Taken together, these results show that the onset of chlorosis during −Fe requires sufficient P in the growth media, and that the "stay-green" phenotype under the combined −Fe−P deficiency is unlikely due to Fe nutritional status in leaves.

**Phosphorus availability modulates the effect of iron deficiency on chlorophyll accumulation and photosystem activity.** To understand the cause of chlorophyll reduction in response to −Fe, we first explored the timing of −Fe sensing and photosynthetic function response. Since −Fe affects chlorophyll accumulation and photosystem II (PSII) activity[15,16], we monitored their kinetics over 172 h (h) (Fig. 2a, b and Supplementary Fig. 2A–C). Arabidopsis plants were first grown on +Fe+P media for 1 week, and then transferred to +Fe+P, −Fe+P, or −Fe−P conditions. −Fe+P caused a significant decrease in chlorophyll content, which was observable starting at 52 h after the transfer to −Fe+P (Fig. 2a). However, transfer to −Fe−P did not affect chlorophyll content, even at 172 h after the transfer (Fig. 2a). To determine how photosynthesis was affected, we measured Fv/Fm, which reflects the quantum yield of photochemistry and is a measure of PSII activity[15,16]. Plants under −Fe+P decreased Fv/Fm, which was observable starting at 52 h, indicative of compromised electron transport through PSII, and which coincides with the decrease of chlorophyll accumulation (Fig. 2b). By 172 h, PSII activity was substantially reduced under −Fe+P compared to +Fe+P. However, plants under −Fe−P showed slightly lower but stabilized Fv/Fm compared to those in +Fe+P (Fig. 2b). Similarly, non-photochemical quenching decayed as a function of time under −Fe+P whereas the decay was substantially less in −Fe−P (Supplementary Fig. 2D). These physiological characterizations showed that chlorophyll accumulation and photosystem activity were affected by −Fe, and both responses were P-dependent (Fig. 2a, b and Supplementary Fig. 2D).

**Phosphorus and iron interaction regulates the expression of photosynthesis-related genes (PRGs).** Based on these findings, we selected three time points at 39, 52, and 76 h after the transfer of plants to +Fe+P, −Fe+P, or −Fe−P to conduct a global gene expression analysis in shoots (Fig. 2c and Supplementary Fig. 3A–F). We identified genes whose expression levels were either increased or decreased by −Fe+P relative to +Fe+P by at least twofold at a $P$ value <0.05 (Fig. 2d and Supplementary Data 1). Even more genes were either upregulated or downregulated in −Fe−P conditions relative to +Fe+P (Fig. 2d and Supplementary Fig. 4A–C). A total of 673 and 2434 transcripts were specifically differentially regulated in response to −Fe+P or −Fe−P, respectively, supporting the existence of different signaling pathways under the two conditions (Supplementary Fig. 4A–C). To identify functions enriched in genes that were differently regulated by −Fe+P or −Fe−P, we performed Gene Ontology (GO) enrichment analysis. The common set of 52 genes that were specifically downregulated by −Fe−P at 52 h and 76 h after the transfer (Supplementary Fig. 5A and Supplementary Data 1) showed enrichment for ribosomal genes (Supplementary Fig. 5B) while upregulated genes (162 genes, Supplementary Fig. 5C, Supplementary Data 1) revealed enrichment for genes involved in cation transport, response to water, and ester hydrolysis (Supplementary Fig. 5D and Supplementary Data 1).

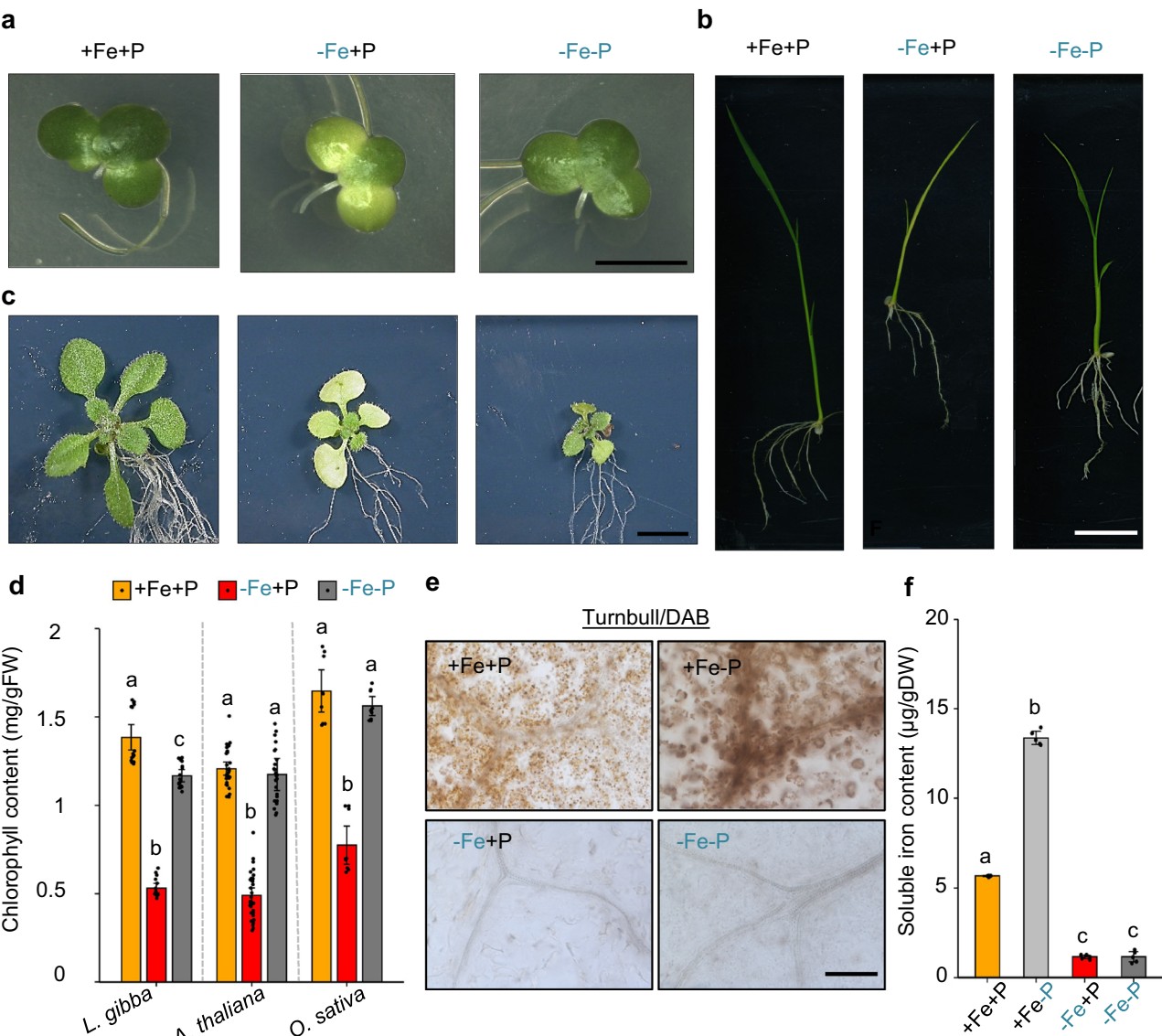

**Fig. 1 Phosphorus deficiency prevents iron deficiency-induced chlorosis in evolutionarily distant plant species. a–c** Duckweed (*Lemna gibba*), rice (*Oryza sativa* cv Nipponbare), and *Arabidopsis thaliana* plants grown on media containing iron and phosphorus (+Fe+P), deficient in iron (−Fe+P), or deficient in both elements (−Fe−P). Representative images of *L. gibba* propagated for 28 days (**a**), 24-day-old rice (**b**) and 14-day-old *A. thaliana* (Col-0) (**c**) are shown. The experiment was repeated with similar results three times. **d** Mean chlorophyll accumulation in *L. gibba*, *A. thaliana*, and *O. sativa* grown under +Fe+P, −Fe+P, and −Fe−P conditions. Scale bars: 7 mm (**a**), 10 mm (**b**), and 5 mm (**c**). Data shown are from three experiments with 3–10 plants per experiment. Error bars represent 95% confidence intervals. FW: fresh weight. **e** Turnbull staining of $Fe^{2+}$ in leaves of seedlings grown on +Fe+P for ten days and transferred to +Fe+P, +Fe−P, −Fe+P or −Fe−P conditions for 7 days. The experiment was repeated with similar results three times. **f** Mean soluble iron content in shoots of fourteen-day-old *A. thaliana* plants grown on agar plates containing +Fe+P, +Fe−P, −Fe+P, or −Fe−P. Data shown are from three experiments with ten plants per experiment. Error bars represent 95% confidence intervals. DW dry weight. For **d** and **f**, letters above bars represent statistically different means within each species-specific comparison at $P < 0.05$ (one-way ANOVA with a Duncan post hoc test). Source data are provided as a Source Data file.

On the other hand, GO analysis of the 32 genes specifically downregulated by −Fe+P but not affected by −Fe−P at 52 h and 76 h (Fig. 2e and Supplementary Data 1) revealed an enrichment of genes related to the chloroplast and photosynthesis-related processes (Fig. 2f), while upregulated genes (Supplementary Fig. 5E) were enriched for genes related to cellular respiration, oxidation–reduction process, and energy metabolism (35 genes; Supplementary Fig. 5F). Altogether, the transcriptomics analysis indicated that the control of chloroplast function is an integral component of the nuclear transcriptomic response to −Fe, which is dependent on P availability. We also learned that the photosynthesis-related phenotypes we observed under −Fe+P, but not under −Fe−P, could be due to the downregulation of key photosynthesis regulators.

**PHT4;4-mediated AsA transport in the chloroplast prevents chlorosis in −Fe−P conditions.** To decode the signaling pathways that control the expression of the photosynthesis genes in response to −Fe+P, we exploited natural variation in expression of the 32 genes that were downregulated by Fe deficiency in a P-dependent manner in a worldwide collection of *A. thaliana* accessions[17]. One way to identify mechanisms regulating the expression of photosynthesis-related genes (PRGs) could be to identify genetic factors associated with their natural variation of

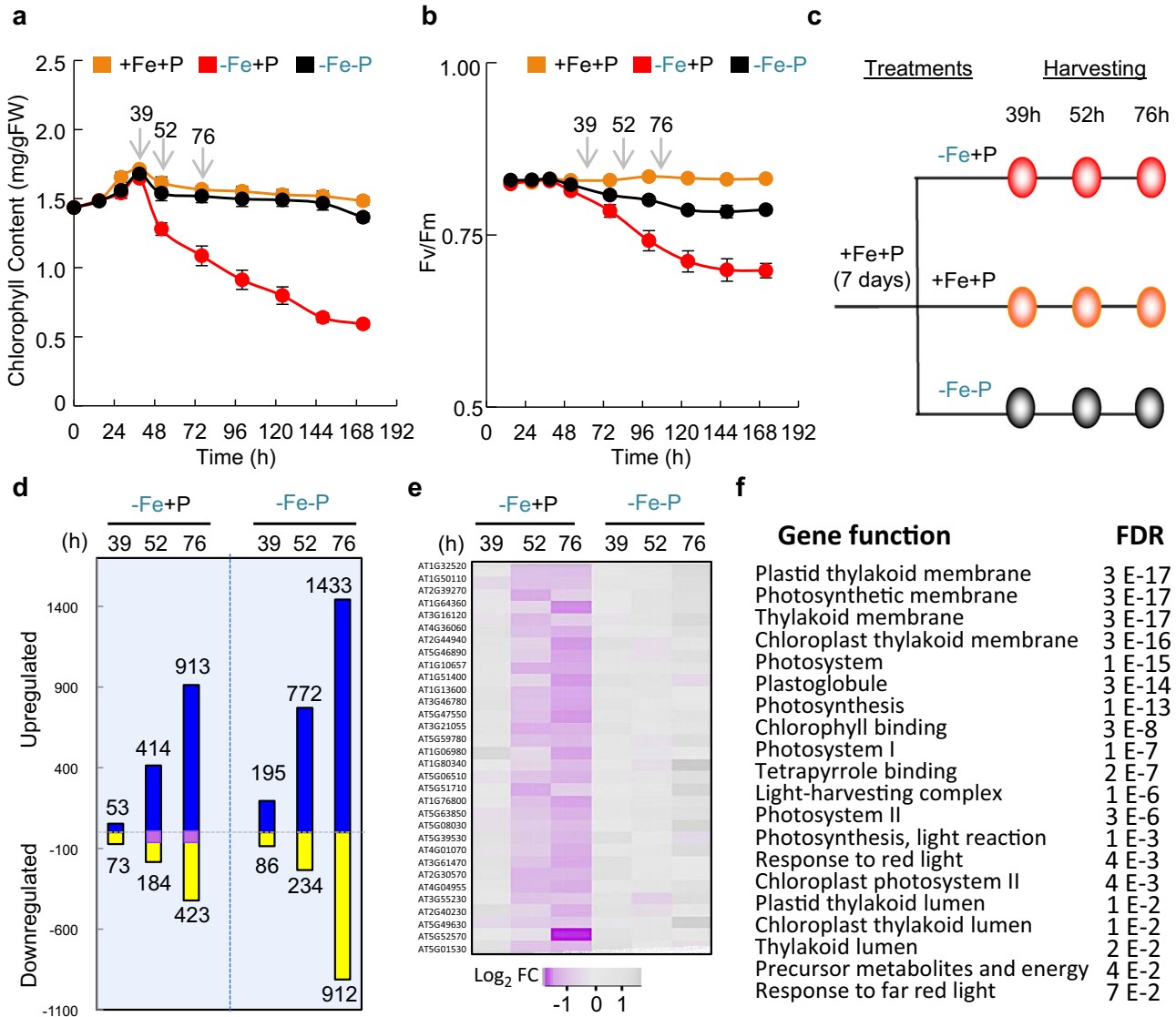

**Fig. 2 Kinetics of chlorophyll accumulation, photosystem II activity, and transcriptome change in response to iron and phosphorus availability. a**, **b** Chlorophyll content and PSII activity (Fv/Fm) in response to iron and/or phosphate deficiency in *A. thaliana*. Seedlings were grown for 7 days in the presence of iron and phosphorus (+Fe+P) and transferred to three different media: +Fe+P, −Fe+P, or −Fe−P for 15, 28, 39, 52, 76, 100, 124, 148, and 172 h. **a** Mean chlorophyll data shown are from three experiments with eight plants per experiment. Error bars represent 95% confidence intervals. FW fresh weight. **b** Mean Fv/Fm data shown are from three experiments with 16 plants per experiment. Error bars represent the 95% confidence intervals. **c** Experimental design for transcriptomic studies on *A. thaliana* (Col-0) shoots. Plants were grown in +Fe+P media for 7 days before transfer to three different media conditions (+Fe+P, −Fe+P, or −Fe−P). Shoots were harvested for RNA extraction and sequencing at the 39, 52, and 76 h time points. **d** Global expression analysis of genes in response to −Fe+P and −Fe−P relative to +Fe+P. Numbers of genes displaying at least twofold change in their expression are shown for each condition. The 32 genes that were decreased specifically in −Fe+P but not in −Fe−P relative to +Fe+P at 52 h and 76 h (highlighted in purple) were used to perform genome-wide association studies. **e** A heatmap showing gene expression patterns of the 32 genes in −Fe+P and −Fe−P relative to control (+Fe+P) at 39, 52, and 76 h after transfer. LogFC log2 fold change. **f** Gene Ontology enrichment for the 32 genes whose mRNA abundance was specifically decreased by −Fe+P. FDR false-discovery rate. Source data are provided as a Source Data file.

expression across Arabidopsis accessions. For this, we retrieved the published mRNA levels of the 32 photosynthesis genes in shoots of 727 Arabidopsis accessions[17]. Strikingly, their expression was generally positively associated with each other across the accessions, indicating that they are co-regulated (Supplementary Fig. 6A). Next, we performed principal component analysis (PCA) to reduce the dimensionality of expression data for these 32 genes. PC1 explained 89.5% of the variation in their expression (Supplementary Fig. 6B), which we then used to perform a genome-wide association study (GWAS) (Fig. 3a). Our GWA analysis detected 38 QTLs containing 145 candidate genes in total (Fig. 3a). In this study, we followed up two QTLs, the first one

located on chromosome 1 (SNP4653399) containing nine candidate genes (AT1G13570 to AT1G13610) (Supplementary Fig. 7), and the second QTL located on chromosome 4 (SNP171674) containing six candidate genes (AT4G00355 to AT4G00400) (Supplementary Fig. 7). All of the candidate genes underlying these two QTLs were subjected to functional genetic analysis to identify the causal gene(s) that influence the expression of the PRGs and chlorophyll content.

Analysis of mutants of genes underlying the QTL for chlorophyll accumulation on chromosome 4 (associated with SNP171674) showed that genetic inactivation of only one gene, *AT4G00370* (*PHT4;4*), failed to stay green under −Fe−P

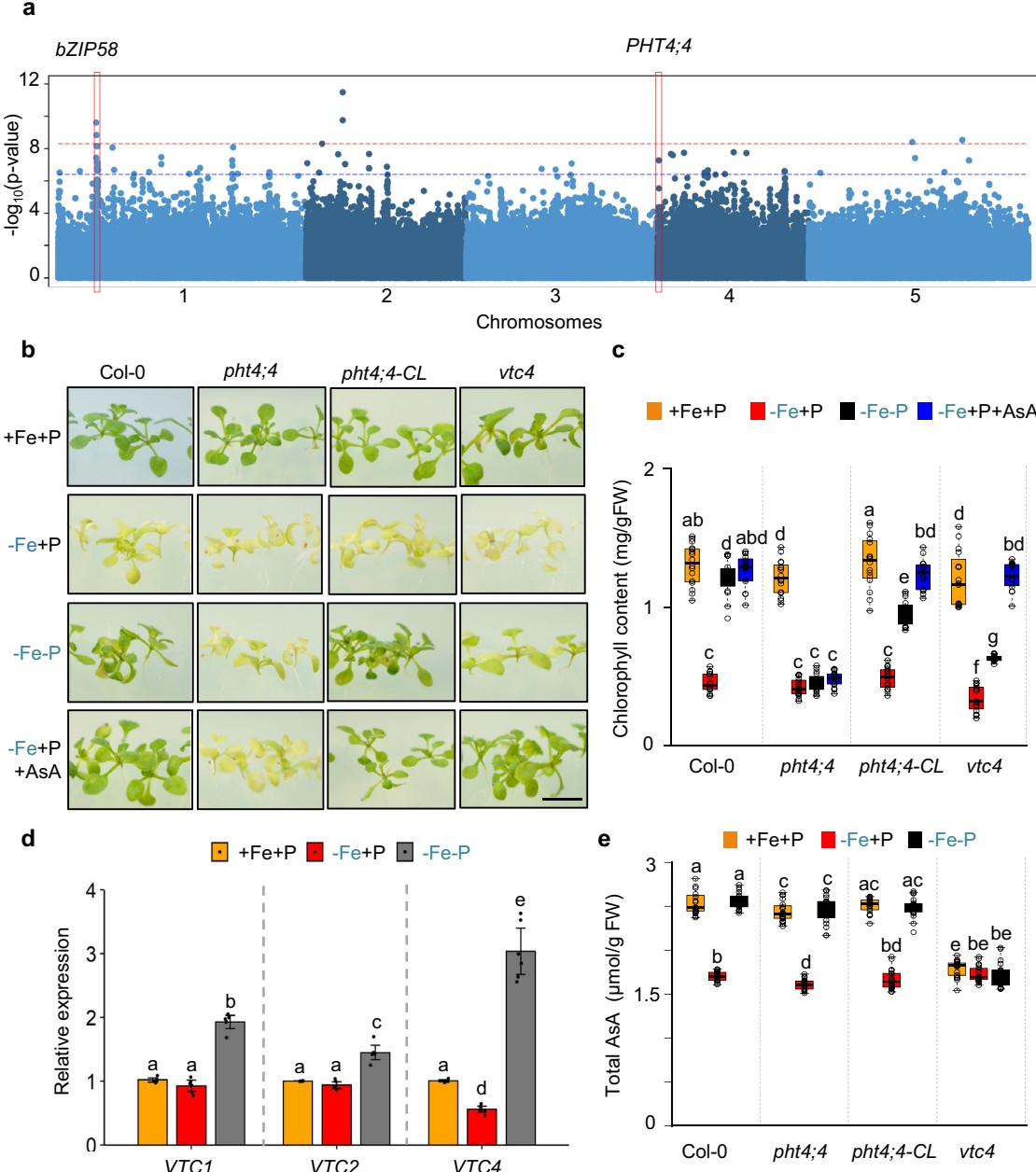

**Fig. 3 PHT4;4 prevents chlorosis under the combined deficiency of iron and phosphorus. a** A Manhattan plot for genome-wide association mapping using principal component 1 that explained 89.5% of expression variation of the 32 photosynthesis-related genes across 727 *A. thaliana* accessions[17]. The five chromosomes are depicted by alternating light- and dark-blue colors. Dashed lines correspond to significant SNPs determined using an accelerated mixed model with adjustment for multiple tests using a FDR 5% threshold (blue) and Bonferroni $\alpha = 0.05$ (red). The light-gray rectangle highlights a significant association located in an intergenic region (SNP: 4493712). Two significant associations that were followed up in this study are highlighted in red rectangles. **b** Representative images of wild-type Col-0, *pht4;4, vtc4*, and a line expressing genomic *PHT4;4* in *pht4;4* (*pht4;4-CL*) grown for 7 days in the presence of iron and phosphorus (+Fe+P) and transferred to three different media: +Fe+P, −Fe+P, or −Fe−P for 7 additional days. Scale bars: 7 mm. The experiment was repeated with similar results three times. **c** Total chlorophyll content in Col-0, *pht4;4, vtc4*, and *PHT4;4-CL* grown under +Fe+P, −Fe+P, −Fe−P, or −Fe+P+AsA. FW: fresh weight. Data shown are from ten plants conducted in three independent experiments. **d** Relative mRNA abundance of *VTC* genes (*VTC1* (AT2G39770), *VTC2* (AT4G26850), and *VTC4* (AT3G02870)) in shoots of Col-0 plants grown on +Fe+P media for 7 days and transferred to +Fe+P, −Fe+P, or −Fe−P media for 52 h. Data shown are the means from 3 experiments. Error bars represent the 95% confidence interval. **e** Total ascorbic acid (AsA) content in Col-0, *pht4;4, vtc4*, and *PHT4;4-CL* plants grown for 7 days on +Fe+P media and transferred to +Fe+P, −Fe+P, or −Fe−P media for 52 h. Data shown are from three experiments, each with 16 plants. In box plots (**c, e**), center lines indicate sample medians; box limits indicate the 25th and 75th percentiles; whiskers extend 1.5 times the interquartile range from the 25th and 75th percentiles. For **c–e**, letters above bars or boxes represent statistically different means at $P < 0.05$ (one-way ANOVA with a Duncan post hoc test). Data points are plotted as open circles. Source data are provided as a Source Data file.

conditions, and exhibited a remarkable decrease in chlorophyll content by comparison to wild-type plants (Fig. 3b, c and Supplementary Fig. 8). Introduction of the wild-type *PHT4;4* allele into a *pht4;4* mutant background complemented these phenotypes (Fig. 3b, c). *PHT4;4* encodes a protein that transports inorganic phosphate (Pi) as well as ascorbic acid (AsA) to the chloroplast[18]. To test whether other chloroplastic Pi transporters[18] influenced chlorophyll accumulation under combined Fe and P deficiency, we characterized mutant plants for *PHT4;3*, *PHT4;5*, and *PHT4;6*. Chlorophyll content was not distinguishable in any of these mutants compared to wild-type plants in −Fe−P conditions (Supplementary Fig. 9A). Furthermore, plants lacking key genes regulating Pi transport or homeostasis, the transcription factor *PHOSPHATE RESPONSE 1* (*PHR1*) and the *SPX DOMAIN GENE 1* (*SPX1*), showed similar chlorophyll content as wild-type plants (Supplementary Fig. 9B). Taken together, these results support the idea that the effects of *pht4;4* mutants on chlorophyll content are likely mediated through its AsA transport activity.

To test our hypothesis about the role of AsA in preventing chlorosis under −Fe−P conditions, we first assessed how Fe and P availabilities regulate the expression of *VITAMIN C* (*VTC*) enzymes involved in AsA biosynthesis in plants[19]. Our RNA-seq analysis indicated that −Fe−P caused a two- to threefold increase in *VTC1* (AT2G39770), *VTC2* (AT4G26850), and *VTC4* (AT3G02870) expression, which we confirmed using qRT-PCR (Fig. 3d and Supplementary Fig. 10A). However, −Fe in the presence of P (−Fe+P) caused about a twofold decrease in the mRNA abundance of *VTC4* (Fig. 3d and Supplementary Fig. 10A). VTC4 is the final enzyme in the AsA biosynthesis pathway[19]. This prompted us to test the effect of the absence of *VTC4* on chlorophyll accumulation under −Fe+P and −Fe−P conditions. Under −Fe−P, mutants with a *vtc4* null allele were still chlorotic, similarly to *pht4;4* and in contrast to wild-type plants (Fig. 3b, c). These data show that AsA contributes to preventing chlorosis in −Fe−P conditions.

Next, we tested whether the chlorotic phenotype is due to variations in AsA levels. In wild-type, AsA levels decreased significantly under −Fe+P at 52 h after the transfer relative to control (+Fe+P), whereas no change was detected under −Fe−P (Fig. 3e), suggesting that AsA levels were associated with −Fe-mediated chlorosis. To test whether AsA levels were associated with chlorosis in general, we measured AsA levels in AsA synthesis (*vtc4*) mutant plants. Under +Fe+P, *vtc4* plants accumulated 35% less AsA than wild-type plants, and AsA levels remained unchanged in response to −Fe+P or −Fe−P stress (Fig. 3e). However, *vtc4* plants did not show the chlorotic phenotype under +Fe+P, which indicated that the level of AsA contributed to the chlorotic phenotype specifically under −Fe and this contribution was dependent on P availability. In addition, the AsA transporter (*pht4;4*) mutants showed similar AsA levels as the wild type even though *pht4;4* plants were still chlorotic in −Fe−P (Fig. 3e). To determine whether AsA accumulation in the cell or its transport to the chloroplast is associated with the development of chlorotic phenotype in −Fe+P, we tested the effect of an exogenous supply of AsA in wild type, *vtc4*, and *pht4;4* plants (Fig. 3b, c). Exogenous AsA alleviated the chlorosis caused by −Fe+P in wild-type and *vtc4* mutant plants. However, *pht4;4* mutants failed to stay green under −Fe+P+AsA conditions (Fig. 3b, c), indicating that the transport of AsA to the chloroplast is required for −P-mediated "stay-green" phenotype under Fe deficiency. Our results showed that −P prevents the downregulation of *VTC4* by −Fe and associated changes in AsA accumulation, and that the PHT4;4-mediated transport of AsA to chloroplasts is required for the maintenance of chlorophyll content under combined deficiency of Fe and P.

We next asked whether *PHT4;4*-mediated AsA transport to the chloroplast is important for regulation of the PRGs that were specifically downregulated by −Fe in a P-dependent manner. First, we tested the effects of *PHT4;4* inactivation on the expression of these photosynthesis-related genes using qRT-PCR (Fig. 4a and Supplementary Fig. 10B, and Supplementary Data 2). While −Fe+P significantly downregulated the mRNA abundance of these genes in wild-type plants (Col-0), −Fe−P prevented this response (Fig. 4a and Supplementary Fig. 10B, and Supplementary Data 2). Furthermore, adding AsA to −Fe+P mimicked −Fe−P response in preventing downregulation of the photosynthesis genes (Fig. 4a, Supplementary Fig. 10B, and Supplementary Data 2). While under −Fe+P PRGs are expressed in *pht4;4* to similar levels to that of wild-type plants, PRGs were downregulated in *pht4;4* plants under −Fe−P as well as −Fe+P supplemented with AsA (Fig. 4a and Supplementary Fig. 10B). Taken together, these data indicate AsA regulates the expression of PRGs in a *pht4;4*-dependent manner.

## bZIP58 regulates the expression of photosynthesis-related genes (PRGs) in a Fe-dependent manner

We next sought to determine how −Fe+P affected expression of the 32 PRGs. To look for potential transcriptional regulators of these genes, we screened candidate genes from the GWAS analysis. We found that *bZIP58* (AT1G13600) (Fig. 3a and Supplementary Figs. 6 and 7), a putative transcription factor, underlies one of the strongest QTL peaks (Fig. 3a). To characterize *bZIP58* further, we first asked whether Fe and P deficiencies affected its expression. Fe limitation strongly downregulated *bZIP58* in shoots, and P limitation alleviated the repression of *bZIP58* caused by Fe deficiency (Fig. 4b and Supplementary Fig. 10A). *bZIP58* expression was not altered by +Fe−P, indicating that P availability alone does not influence *bZIP58* expression (Supplementary Fig. 10C). In comparison to shoots, we found a much weaker downregulation of *bZIP58* by −Fe in roots (Supplementary Fig. 10D). Given that AsA supplementation prevented the development of chlorosis caused by Fe limitation, we next asked how AsA would influence the expression of *bZIP58*, and whether bZIP58 regulates the expression of *VTC4* (Fig. 4c). Remarkably, AsA prevented the downregulation of *bZIP58* by −Fe+P. This led us to examine the contribution of bZIP58 in regulating the −Fe+P-specific PRGs under +Fe+P, −Fe+P, and −Fe−P conditions (Fig. 4a, Supplementary Fig. 10B, and Supplementary Data 2). Mutants with the *bzip58* null allele showed a remarkable constitutive decrease in the expression of these 32 PRGs (Fig. 4a, Supplementary Fig. 10B, and Supplementary Data 2). bZIP58 localizes to the nucleus (Fig. 4d), which is consistent with a role as a transcription factor. Furthermore, to ask whether bZIP58 can directly regulate any of the 32 PRGs, we performed transactivation assays using bZIP58 as an effector and the promoters of five randomly selected PRGs (*PHOTOSYSTEM II SUBUNIT T* (*PSBTN*); *PHOTO SYSTEM II 5 kD*, *PLASTID TRANSCRIPTIONALLY ACTIVE 16* (*PTAC16*), *LIGHT HARVESTING COMPLEX PHOTOSYSTEM II* (*LHCB4.1*) and *PHOTOSYSTEM II REACTION CENTER W* (*PSBW*)) fused to GUS as reporters (Supplementary Fig. 11A) showed that bZIP58 can activate the promoter activity of three of the five genes tested (*PSBTN*, *PHOTOSYSTEM II 5 kD*, and *PTAC16*) (Supplementary Fig. 11B). Therefore, bZIP58 can directly activate the expression of some PRGs. These results demonstrate that bZIP58 is a key gene to regulate the expression of PRGs in a Fe-dependent manner.

Taken together, these findings support the idea that bZIP58 is a key regulator of PRGs, and its absence could alter chlorophyll accumulation regardless of Fe and P availability. Genetic inactivation of *bZIP58* indeed causes a constitutive decrease in

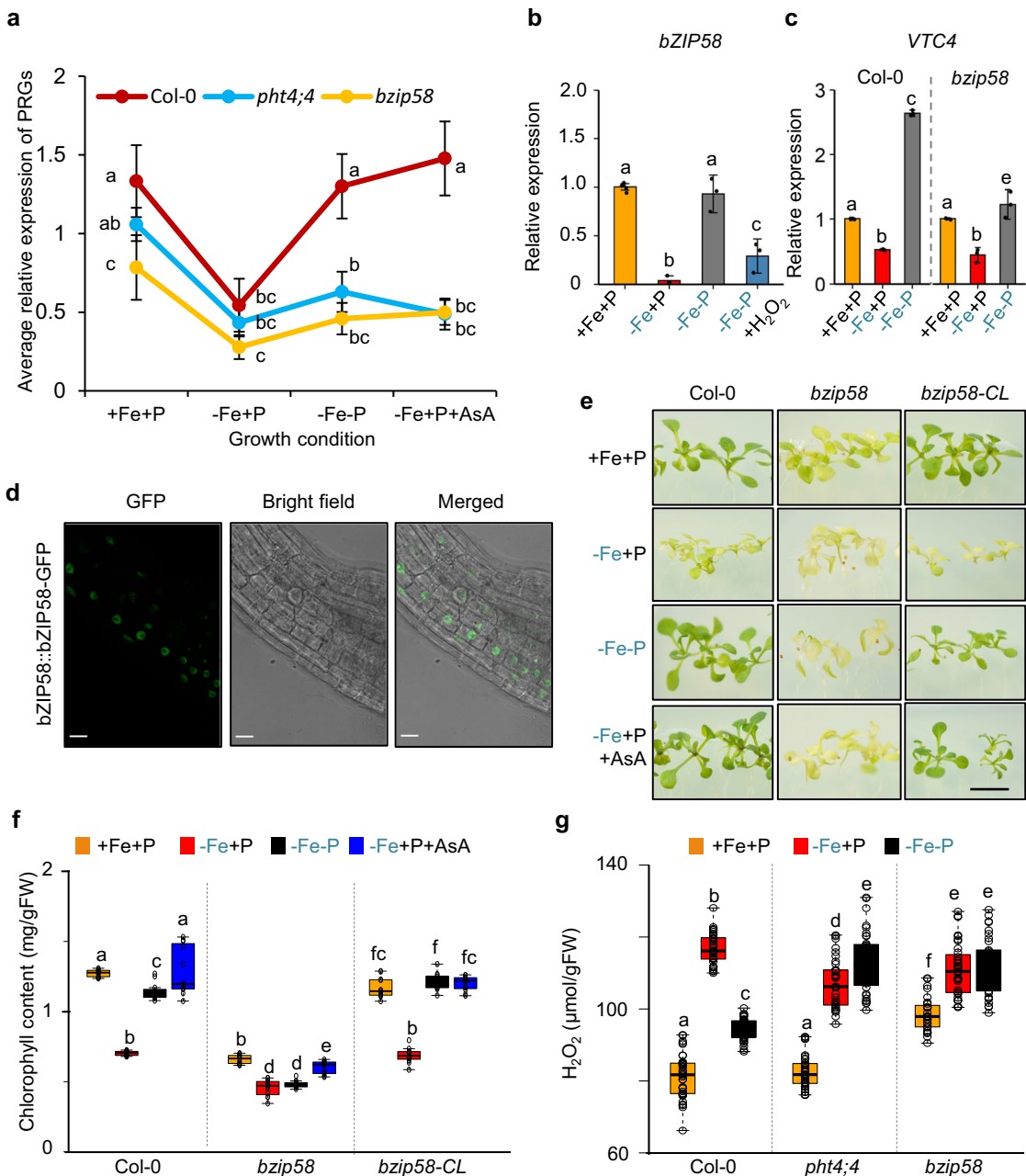

**Fig. 4 bZIP58 regulates photosynthesis-related genes and chlorophyll accumulation. a** Average relative expression of 32 PRGs in Col-0 (red), *pht4;4* (light blue), and *bzip58* (brown) in plants grown for 7 days in the presence of iron and phosphorus (+Fe+P) and transferred to +Fe+P, −Fe+P, −Fe−P, or −Fe−P + AsA for 76 h. Letters above markers represent statistically different means at $P < 0.05$ (one-way ANOVA with a Duncan post hoc test). Error bars = 95% CI. Data for all PRGs were averaged from three independent experiments. Relative expression level of PRGs was determined relative to *Ubiquitin 10*. **b** Relative mRNA abundance of *bZIP58* in shoots of Col-0 plants grown for 7 days on +Fe+P and transferred to +Fe+P, −Fe+P, −Fe−P, or −Fe−P+$H_2O_2$ for 76 h. Data shown are the means from three experiments. Error bars represent 95% confidence intervals. **c** Relative mRNA abundance of *VTC4* in the shoots of Col-0 and *bzip58* mutants grown in the presence of +Fe+P for 7 days and transferred to +Fe+P, −Fe+P, or −Fe−P for 76 h. Data shown are the means three experiments. Error bars represent 95% confidence intervals. **d** Confocal microscopy images of *p35S::bZIP58::GFP* expressing plants grown for 7 days under +Fe+P. Scale bars: 20 μm. The experiment was repeated with similar results three times. **e** Representative images of Col-0, *bzip58*, and a line expressing genomic *bZIP58* in *bzip58* mutants (*bzip58-CL*) grown for 7 days in +Fe+P and transferred to +Fe+P, −Fe+P, −Fe−P, or −Fe +P+AsA for 7 additional days. Scale bars: 7 mm. The experiment was repeated with similar results three times. **f** Total chlorophyll content in Col-0, *bzip58*, and *bZIP58-CL* plants grown for 7 days in +Fe+P and transferred to +Fe+P, −Fe+P, −Fe−P, or −Fe+P+AsA for 7 days. FW: fresh weight. Data shown are from four experiments. **g** Accumulation of $H_2O_2$ (a type of ROS) in shoots of Col-0, *pht4;4* and *bzip58* plants grown for 7 days in +Fe+P and transferred to +Fe+P, −Fe+P, or −Fe−P for 52 h. Data shown from 12 experiments. In box plots (**f**, **g**), center lines show sample medians; box limits indicate the 25th and 75th percentiles; whiskers extend 1.5 times the interquartile range from the 25th and 75th percentiles. For **b**, **c**, **f**, and **g**, letters above bars or boxes represent statistically different means at $P < 0.05$ (one-way ANOVA with a Duncan post hoc test). Data points are plotted as open circles. Source data are provided as a Source Data file.

chlorophyll content, and the mutant line is chlorotic (Fig. 4e, f). The expression of the *bZIP58* gene in *bzip58* plants complements the constitutive chlorosis phenotype, and the complemented line responds to Fe and P availability similarly to wild-type plants (Fig. 4e, f). Furthermore, AsA supplementation could not rescue the chlorotic phenotype of *bzip58* mutants (Fig. 4e, f), indicating that bZIP58 lies downstream of AsA action. In addition, bZIP58 was partially required to induce *VTC4* expression under −Fe−P conditions (Fig. 4c). These data show that bZIP58 controls the expression of PRGs and is transcriptionally regulated in response to −Fe depending on P availability, likely by mediating the perception of AsA.

**Plastid ROS influences chlorophyll content and PRGs expression in −Fe+P conditions**. We next asked how AsA in the chloroplast could affect photosynthesis. AsA has an antioxidizing action that detoxifies reactive oxygen species (ROS) through its scavenging properties[20], thus making ROS a potential signaling molecule[21–23] capable of modulating the expression of PRGs through bZIP58. To test this hypothesis, we first measured the relative amount of ROS accumulation in shoots of wild-type, *pht4;4*, and *bzip58* plants under various Fe and P availability. −Fe +P caused a twofold increase in ROS accumulation in shoots of wild-type plants, which partially depended on P availability (Fig. 4g). *pht4;4* plants displayed comparable ROS accumulation to that of the wild-type under +Fe+P and −Fe+P. However, *pht4;4* plants accumulated significantly higher ROS than wild-type plants under −Fe−P (Fig. 4g). In addition, *bzip58* mutant plants displayed a constitutive increase in ROS accumulation (Fig. 4g). To check whether ROS in turn can regulate the expression of *bZIP58*, we quantified *bZIP58* expression in response to foliar application of $H_2O_2$. ROS treatment caused a 4-fold decrease in *bZIP58* transcript accumulation, and this effect was independent of the availability of Fe or P in the medium (Fig. 4b and Supplementary Fig. 10C). Our results supported the role of ROS as a plastid signal to influence chlorophyll content and PRGs expression in −Fe+P conditions. To test this idea, we expressed *bZIP58* in *bzip58* (Line A1) and *bzip58xpht4;4* mutant plants (Line B1) under the control of a ROS-responsive promoter of a gene called *JUNGBRUNNEN1* (*JUB1*)[24] (Fig. 5a). −Fe induced the expression of *JUB1p::bZIP58* by 2.5-fold in *bzip58* and by fourfold in *bzip58xpht4;4* (Fig. 5a). While WT and bzip58-CL lines were chlorotic and showed repression of PRGs in −Fe+P conditions (Fig. 4e, f), the expression of *JUB1p::bZIP58* enhanced chlorophyll accumulation and expression of PRGs in the *bzip58* mutant (Fig. 5b, c and Supplementary Fig. 12). These responses were exacerbated in *bzip58xpht4;4* double mutant expressing *JUB1p::bZIP58* (Fig. 5b, c and Supplementary Fig. 12). Greatly enhanced induction of *JUB1p::bZIP58* by −Fe in *bzip58xpht4;4* compared to *bzip58* indicates the sensitivity of this ROS-responsive promoter to PHT4;4-dependent plastid signal. Remarkably, AsA supplementation in −Fe+P prevented the induction of *JUB1p::bZIP58* in the *bzip58* mutant but not in *bzip58xpht4;4* double mutant (Fig. 5b, c and Supplementary Fig. 12). Consistently, Line A1 was chlorotic, which was associated with repression of PRGs, and Line B1 accumulated chlorophyll under −Fe+P+AsA conditions and exhibited high expression levels of PRGs (Fig. 5b–d and Supplementary Fig. 12). These results demonstrate that PHT4;4-mediated AsA transport to the chloroplast regulates the expression PRGs in −Fe+P through stress-induced ROS in the chloroplast that is perceived by bZIP58.

## Discussion
Fe is an important micronutrient that plays crucial roles in plant growth and development. In nature, plants may experience Fe deficiency due to low Fe supply and alkaline pH (e.g., pH greater than 7.0 makes Fe unavailable to plants)[25,26]. Fe deficiency causes chlorosis[27,28] and affects root growth[29]. Great progress has been achieved in understanding how plants respond to Fe deficiency, particularly the molecular mechanisms for Fe uptake in the roots and its distribution throughout the plant[30]. However, how plants sense and respond to Fe deficiency to control photosynthesis remains poorly understood. Fe is an essential cofactor in photosystem complexes and is present in almost all the components of the electron transport chain in the chloroplast[6]. Here, we describe a signaling pathway orchestrating communication between the chloroplast and the nucleus to control the expression of photosynthesis-related genes (PRGs) and chlorosis in Fe-deficient environments (Fig. 6). Modulation of this newly discovered pathway could have a direct impact on plant growth in the field by improving plant photosynthetic activity while reducing nutrient supply.

Generally, it has been assumed that chlorosis in Fe-deficient environments is due to Fe deficiency in leaves[26]. However, optimal photosynthesis in plants subjected to simultaneous Fe and P deficiency, as described in this study, challenges this assumption. There are two possibilities to explain the absence of chlorosis under −Fe−P conditions. First, Fe and P interactions can influence their bioavailability, and a joint deficiency of Fe and P may increase bioavailable Fe for photosynthesis. However, we did not find evidence to suggest that levels of bioavailable Fe change in the leaves of plants grown under −Fe−P using our methods; more sensitive methods of quantification with higher spatial resolution would be needed to further reinforce our conclusion. A second possibility is that P availability modulates Fe-deficiency signaling pathways that control photosynthesis. This hypothesis is clearly supported by our global gene expression analysis, which revealed that the downregulation of 32 PRGs by −Fe is prevented by simultaneous Fe and P deficiency. A similar transcriptional response was observed for bZIP58, a key transcriptional regulator of PRG expression we discovered in this study.

Fe and P are known to interact to regulate root growth[13,31]. However, our understanding of the molecular basis of Fe and P signaling crosstalk to regulate photosynthesis remains fragmentary. Therefore, identifying key molecular mechanisms improving plant photosynthesis under nutrient-limited conditions is of great importance. Using expression GWAS, we found that a chloroplastic Pi and AsA transporter PHT4;4[18] controls Fe-deficiency-induced chlorosis. It is noteworthy that our results show that *PHT4;4* expression is not affected by Fe deficiency (Supplementary Fig. 10A). Given its dual-transport activity, one could ask which target of transport, Pi or AsA, is the cause of the persistent chlorotic phenotype in the pht4;4 mutant under the −Fe−P treatment. As far as Pi transport is concerned, our results show that genetic inactivation of other plastid Pi transporters (PHT4;3, PHT4;5, and PHT4;6)[32] or key Pi signaling pathway components (SPX1 and PHR1)[33] respond similarly to wild-type plants under −Fe+P or −Fe−P conditions. Our findings thus support the idea that PHT4;4 affects photosynthesis through its AsA transport activity. In line with this, AsA supplementation fully prevents the development of chlorosis in wild-type plants and not in *pht4;4* plants, indicating that this phenotype depends on PHT4;4-mediated AsA transport to the chloroplast.

In our study, we measured AsA content 52 h after Fe-deficiency treatment, and we found a significant decrease in AsA accumulation in leaves. AsA levels have been shown to increase in response to 2–3 weeks Fe-deficiency treatments in *Brassica napus* leaves and sugar beet roots[34,35]. However, 3–7 days Fe-deficiency treatment in Arabidopsis exhibited a slight reduction in AsA accumulation in shoots[36]. It is intriguing to find a consistent reduction of AsA during the early phase of Fe-

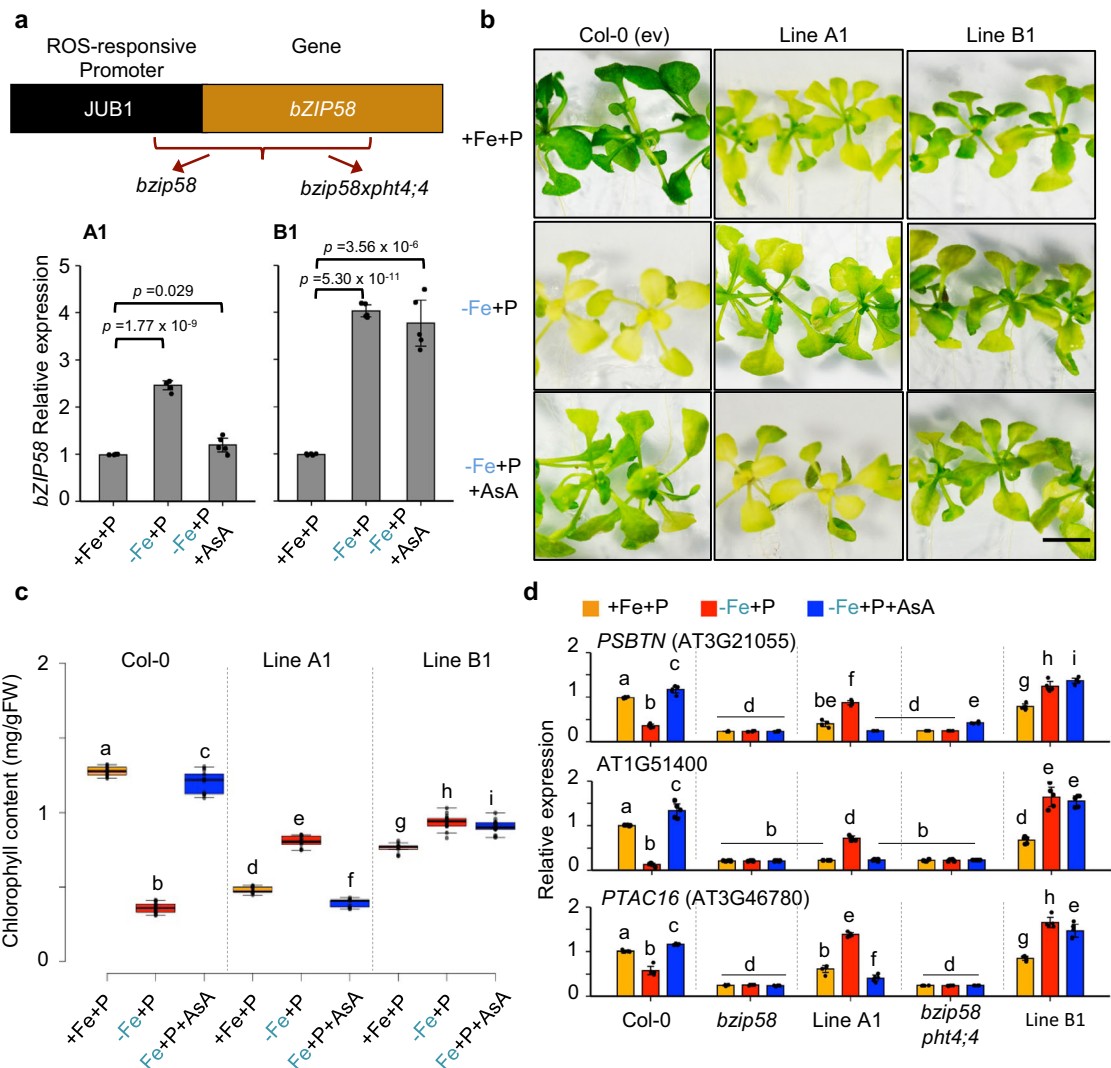

**Fig. 5 Plastid ROS influences chlorophyll content and PRGs expression in a Fe-dependent manner. a** Relative mRNA abundance of *bZIP58* in shoots of plants expressing *JUNGBRUNNEN1p::bZIP58* in the *bzip58* (Line A1) and *bzip58xpht4;4* (Line B1) mutant backgrounds. Plants were grown in the presence of iron and phosphorus (+Fe+P) for 7 days and transferred to +Fe+P, −Fe+P, or −Fe+P+AsA for 76 h. Data shown are the means from five biologically independent samples. Error bars indicate 95% confidence intervals. Asterisks indicate statistically different *bZIP58* expression in the treatment compared to the control (+Fe+P) (*P* value < 0.01, two-sided Student's *t* test). **b** Representative images of wild-type Col-0, Line A1 and Line B1 grown for 1 week under +Fe+P and transferred to three different media conditions (+Fe+P, −Fe+P, or −Fe+P+AsA) for 2 additional weeks. Scale bars: 7 mm. The experiment was repeated with similar results three times. **c** Total chlorophyll content in Col-0, Line A1 and Line B1 grown under +Fe+P, −Fe+P, or −Fe+P+AsA. FW: fresh weight. Data shown are from ten plants representing three independent experiments. Center lines show the medians; box limits indicate the 25th and 75th percentiles; whiskers extend 1.5 times the interquartile range from the 25th and 75th percentiles. **d** Relative mRNA abundance of PRGs genes (*PSBTN* (AT3G21055), AT1G5140, and *PTAC16* (AT3G46780)) in shoots of Col-0, *bzip58*, Line A1, *bzip58 pht4;4*, and Line B1. Plants were grown under +Fe+P for 7 days and transferred to +Fe+P, −Fe+P, or −Fe+P+AsA for 76 h. Data shown are the means from five biologically independent samples. Error bars indicate 95% confidence intervals. For **c**, **d**, letters above bars or boxes represent statistically different means at *P* < 0.05 (one-way ANOVA with a Duncan post hoc test).

deficiency. One possibility is that AsA reduction could be one of the earliest signs of Fe-deficiency stress. Supporting this hypothesis is a recent finding that an early response to salt stress is repression of AsA biosynthesis by ABI4-mediated repression of VTC2[37]. Identifying the upstream factors of the signaling pathway we discovered, including AsA biosynthesis, will be an important future direction.

Fe deficiency causes a decrease in electron flux in the thylakoid membranes[38], and leads to an increase of ROS compared to +Fe conditions[39,40]. Several abiotic stresses enhance ROS accumulation in the chloroplast, a major site of ROS production[22,41,42]. Recently, ROS was proposed to act as a retrograde signal to

influence plant responses to abiotic stresses, particularly for plant tolerance to salt stress[43]. This raises an exciting question of whether chloroplastic ROS could act as a retrograde signal under Fe deficiency to regulate the expression of nuclear PRGs. Our study indeed shows that, under −Fe conditions, ROS acts as a plastidic signal to influence the expression of PRGs and chlorophyll content via the repression of *bZIP58*. This effect becomes clearer when we express bZIP58 under the control of the ROS-inducible promoter of *JUNGBRUNNEN1 (JUB1)*[24]. The induction of *JUB1p::bZIP58* by −Fe+P enhanced chlorophyll accumulation and expression of PRGs in the *bzip58* mutant, and these responses were exacerbated in *bzip58xpht4;4* double mutant.

**Fig. 6 A schematic model delineating a signaling pathway that integrates Fe and P availability cues to regulate chlorophyll accumulation and photosynthesis genes.** Fe deficiency (−Fe+P) causes a decrease in the expression of *bZIP58* that is central to controlling the transcription of nuclear-encoded photosynthetic genes. P limitation under Fe deficiency (−Fe−P) prevents this downregulation of *bZIP58* and induces *VTC4*. The induction of *VTC4* expression requires bZIP58, whose effect could be direct or indirect, represented here by "X". We propose that induction of *VTC4* increases ascorbic acid in the chloroplast mediated by PHT4;4. We hypothesize that the increase of ascorbic acid level prevents ROS accumulation, thus maintaining the expression of bZIP58 and its downstream photosynthesis genes and leading to the "stay-green" phenotype. The figure was created with BioRender.com.

These new results demonstrate that Fe-deficiency-based chlorosis and PRG repression is mediated by chloroplastic ROS controlling the expression of the nuclear transcription factor bZIP58.

Taken together, this study identifies and validates a signaling pathway involved in the regulation of photosynthesis under combined Fe and P stresses (Fig. 6). This newly identified mechanism includes genes encoding chloroplastic (PHT4;4) and nuclear (bZIP58) proteins that prevent the repression of core photosynthesis genes and associated chlorosis under Fe and P co-deficiency. Furthermore, we provide evidence that bZIP58 regulates the expression of AsA biosynthesis genes and, using *pht4;4* mutant plants, we reveal that AsA transport into the chloroplast is important for preventing chlorosis under Fe and P co-deficiency. We also show that in −Fe conditions, the increase of plastidic ROS levels is perceived at the nucleus as a signal to downregulate the expression of *bZIP58* and downstream photosynthesis genes, which results in chlorosis. These results provide fundamental new insights into chlorophyll accumulation and photosynthesis under Fe limitation and identify a signaling pathway that may coordinate plastid–nuclear communication as a means to adapt photosynthesis to nutrient availability.

## Methods
**Plants and growth conditions.** Seeds of *Arabidopsis thaliana* wild-type (ecotype Columbia, Col-0, CS60000) and knockout mutant lines SALK_139877 (AT1G13570), SALK_150849 (AT1G13580), SALK_063177 (AT1G13590), N571881 (AT1G13600), SALK_087271 (AT1G13605), SALK_130208 (AT1G13607), SALK_023173 (AT1G13608), SAIL_1243_E04 (AT1G13609), SAIL_897_D11 (AT1G13610), N469134 (At4g00355), SALK_128714 (AT4G00360), N469134 (AT4G00370), SAIL_842_E09 (AT4G00380), N866595 (At4g00390), SAIL_633_E10 (At4g00400), SALK_077222 (AT3G02870) SALK_053900 (AT3G46980), SALK_114708 (AT5G20380), SAIL_809_B01 (AT5G44370), SALK_067629 (AT4G28610), and SALK_039445 (AT5G20150) were obtained from the Nottingham Arabidopsis Stock Centre (NASC). Homozygous mutant lines were confirmed by PCR using the primers listed in Supplementary Table 1. bZIP58 complemented lines (bZIP58-CL) were generated by expressing 3896 bp genomic DNA containing *bZIP58* in the *bzip58* mutant background (NASC, N571881). Complementation of *pht4;4* mutant plants (PHT4;4-CL) was obtained by expressing 6450 bp genomic DNA containing *PHT4;4* in the *pht4;4* mutant background (NASC, N469134). The *bzip58xpht4;4* double mutant was obtained through the genetic crossing. Arabidopsis plants were grown on control (+Fe+P) plates containing 1.249 mM $KH_2PO_4$; 0.25 mM $Ca(NO_3)_2$; 0.5 mM $KNO_3$; 1 mM $MgSO_4$; 100 µM $FeSO_4.7H_2O$; 30 µM $H_3BO_3$; 1 µM $ZnCl_2$; 10 µM $MnCl_2$; 1 µM $CuCl_2$; 0.1 µM $(NH_4)_6Mo_7O_{24}$; and 50 µM KCl; 0.05% 2-(N-morpholino)ethanesulfonic acid (MES), without sucrose supplementation, and 0.8% washed agar. The agar was washed three times with 50 mM EDTA, pH 5.7, with continuous stirring for 16 h, then washed 6 times with Milli-Q de-ionized water for 2 h to reduce mineral contamination[44]. P-deficient media contained 12.49 µM $KH_2PO_4$ (+Fe−). Fe-free media was obtained by omitting $FeSO_4.7H_2O$ from the growth media (−Fe+P). P- and Fe-deficient media contained 12.49 µM $KH_2PO_4$ (+Fe−), and no $FeSO_4.7H_2O$ (−Fe−P). Seeds were stratified at 4 °C for

3 days and grown on vertical agar plates in a growth chamber with 22 °C, 24 h of light at 100 µmol m⁻² s⁻¹ fluorescent illumination. *Lemna gibba* (duckweed) plants used during this study were obtained from Duckweeds stock center (stock number 29-DWC131) at Rutgers University (USA). Duckweed plants were grown in 1× Schenk & Hildebrandt (SH) hydroponic medium containing 0.05% 2-(N-morpholino)ethanesulfonic acid (MES) and 1% sucrose, and pH adjusted to 5.7. For experiments with duckweeds, P-deficient and Fe-deficient media contained 1% $NH_4H_2PO4$ and 1% $FeSO_47H_2O$, respectively, of 1X SH media. Media were changed every 7 days. The growth condition was 22 °C and 24 h of light at 80 µmol m⁻² s⁻¹. Rice (*Oryza sativa* cv Nipponbare) plants were grown hydroponically in 0.25X Yoshida media[45] under light/dark cycle of 14/10 h, and temperature of 28/25 °C. Single (−P or −Fe) and combined (−P−Fe) nutrient deficiency stresses were applied to 10 day-old plants. $NaH_2PO_4$ (0.33 mM) and Fe-NaEDTA (0.04 mM) present in the complete media were omitted in the P- and/or Fe-deficient media.

**Plasmid construction and plant transformation.** The *JUNGBRUNNEN1* (*JUB1*)[24] promoter and The *bZIP58* coding region were amplified using PCR using primers listed in Supplementary Table 1. The PCR product was and then cloned into the binary vector pCAMBIA1301 using PstI and BamHI restriction enzymes. The PstI cut site was used for the insertion of the *JUB1* promoter upstream of the *bZIP58* coding region. The constructs were transformed into *Agrobacterium tumefaciens* strain GV3101 and then used for Arabidopsis transformation via the floral dip method[46]. Transgenic plants were selected by antibiotic resistance, and homozygous descendants of hemizygous T2 plants segregating 3:1 for antibiotic resistance:sensitivity were used for analysis.

**Iron concentration measurement.** Arabidopsis seeds were germinated and grown in the control (+Fe+P) media for 7 days, and then transferred to +Fe+P, iron-deficient (−Fe+P), phosphate-deficient (+Fe−), or iron and phosphate-deficient (−Fe−P) conditions and grown for 7 additional days. Plants were harvested and shoot samples were dried at 70 °C for 3 days. Total iron was extracted by acid digestion in 1N nitric acid using MARSX (CEM) microwave digester. A 1:10 dilution of the digested material was used to quantify total iron with inductively coupled plasma atomic emission spectrometry (ICP-OES).

**Analysis of photosystem II activity.** Photosystem II (PSII) activity was defined as the maximum quantum yield of the primary quinone acceptor PSII, which was estimated by the ratio of variable fluorescence (Fv) and maximal fluorescence (Fm) of the chlorophyll, Fv/Fm[47]. Arabidopsis wild-type (Col-0) seeds were germinated and grown in control (+Fe+P) for 7 days then transferred to three different media: +Fe+P, iron-deficient (−Fe+P), and iron- and phosphate-deficient (−Fe−P) conditions for 0 h (time of the transfer), 15 h, 28 h, 39 h, 52 h, 76 h, 100 h, 124 h, 148 h, and 172 h. Plates containing the seedlings were dark-adapted for 30 min followed by a very short (160 µs) exposure to a blue measuring beam to determine the minimal fluorescence (F0). The intensity of the detecting and the continuous illumination used was of 156 µE m⁻² s⁻¹. A saturating light flash (2600 µE m⁻² s⁻¹, 250 ms) was applied to measure the maximum fluorescence (Fm). Kinetics were normalized to the maximum fluorescence (Fm). The maximum quantum yield of Photosystem II (Fv/Fm = (Fm − F0)/Fm) was measured for each growth condition, and non-photochemical quenching was calculated using (Fm/Fm') − 1[47].

**Chlorophyll content measurement.** Seeds of Arabidopsis genotypes were germinated and grown in control (+Fe+P) media for 7 days then transferred to three different media: +Fe+P, iron-deficient (−Fe+P), and iron- and phosphate-

deficient (−Fe−P) conditions. Fresh leaves (~30 mg) were incubated in 2.5 mL of 80% acetone overnight in the dark at 4 °C. Total chlorophyll content was measured using a UV–VIS spectrophotometer (Beckman Coulter, DU 530). The absorbance of the supernatant was measured at 645 and 633 nm. The concentration of total chlorophyll was calculated using the following equation[48]: 20.31 $A_{645}$ + 8.05 $A_{663}$/ FW [µg g$^{-1}$] (FW: fresh weight of tissue in grams).

**Ascorbic acid content determination.** Seeds of Arabidopsis genotypes were germinated and grown in control (+Fe+P) media and then transferred to +Fe+P, −Fe+P, or −Fe−P media for 76 h. Ascorbic acid (AsA) content was measured by a colorimetric assay[49]. Briefly, shoots were collected and homogenized in ice-cold 6% trichloroacetic acid (TCA) (Sigma-Aldrich). The homogenate was centrifuged at 6000 rpm for 25 min at 4 °C, and the supernatant was collected. In the supernatant, $Fe^{3+}$ (ferric ion) is reduced by AsA to the $Fe^{2+}$ (ferrous ion) that, when coupled with 2,2-dipyridyl, forms a complex with a characteristic absorbance at 525 nm[49]. A standard curve was generated using known concentrations of AsA made in 6% TCA to determine the AsA concentration. Blanks were prepared using only 6% TCA. AsA concentration was expressed as µmol g$^{-1}$ fresh weight.

**Hydrogen peroxide quantification.** Seeds of Arabidopsis genotypes were germinated and grown in control (+Fe+P) media and then transferred to +Fe+P, −Fe+P, or −Fe−P media for 76 h. Hydrogen peroxide ($H_2O_2$) (Sigma-Aldrich) was quantified as described previously[50,51]. Fresh shoot tissues (0.2 g) were homogenized with 0.1% (w/v) TCA and were centrifuged at 12,000× $g$ for 15 min at 4 °C. In total, 0.5 ml of supernatant was added to 0.5 ml of 10 mM potassium phosphate buffer (pH 7.0) and 1 ml of 1 M potassium iodide. The absorbance of the reaction mixture was measured at 390 nm. The amount of $H_2O_2$ was calculated using a standard curve prepared from known concentrations of $H_2O_2$ ranging from 0.1 to 1 mM.

**RNA sequencing and analysis.** Arabidopsis wild-type (Col-0) plants were grown in control (+Fe+P) media for 7 days and transferred to three different media: control (+Fe+P), iron deficiency (−Fe+P), and iron and phosphate deficiency (−Fe−P) conditions. Shoots were collected at 39, 52, and 76 h after the transfer. For RNA-seq experiments, three biological replicates were prepared for each time point (39, 52, and 76 h) and each condition (+Fe+P, −Fe+P, and −Fe−P) for a total of 27 samples. Total RNA was extracted from these samples using RNeasy Plant Mini Kit (QIAGEN) using the RLT buffer supplemented with 2-mercaptoethanol, and RNA quality was verified using an Agilent 2100 BioAnalyzer. The mRNAs were subsequently isolated using magnetic KAPA Biosystems oligo-dT beads from KAPA Biosystems (Roche) and then used for library construction using the KAPA Biosystems RNA HyperPrep Kit (Roche). To index the libraries, we used adapters from the KAPA Biosystems Single-Indexed Adapter Set A + B (Roche). Before pooling the libraries, we monitored their quality and concentrations using an Agilent 2100 BioAnalyzer, Qubit dsDNA HS Assay Kit (Thermo Fisher Scientific), and the KAPA Library Quantification Kit (Roche). Pooled libraries were then sequenced using the NextSeq 500 System at the Stanford Functional Genomics Facility (Stanford, CA). Raw reads were demultiplexed and aligned to the TAIR10 genome annotation using HISAT2[52] on the Galaxy web platform[53]. Finally, mapped read counts were used to perform normalization and differential expression analysis on R using the DESeq2[54] and TxDB.Athaliana.BioMart.plantsmart[27] (Bioconductor) packages. In DESeq2, $P$ values from the Wald test were corrected for multiple hypothesis testing using the Benjamini and Hochberg method. A transcript was considered differentially expressed if the adjusted $P$ value <0.05. Volcano plots were generated using the EnhancedVolcano package (version 1.6.0) (Bioconductor) with a default cut-off of log2(fold change) >|2| and adjusted $P$ value <10e$^{-6}$. DEGs having a $P$ value of 0 were converted to $10^{-1}$ × lowest nonzero $P$ value.

**Turnbull staining.** Arabidopsis plants (Col-0 genotype) were grown on 0.5× MS medium. Fourteen days after germination, plants were vacuum infiltrated with Turnbull staining solution (4% (v/v) HCl and 4% (w/v) K-ferricyanide) for 15 min and then incubated for 45 min at room temperature on an orbital shaker[55,56]. To inhibit peroxidases and redox reactions, the plants were washed with distilled water and treated for 1 h in methanol containing 0.065% (w/v) sodium azide and 0.3% (v/v) $H_2O_2$. After washing with 0.1 M potassium phosphate buffer (pH 7.4) followed by washing with sterile water, the plants were placed on a slide and observed under a microscope (Olympus BX61).

**Real-time quantitative reverse-transcription PCR.** Seeds of Arabidopsis wild type (Col-0), bzip58, and pht4;4 mutant plants were germinated and grown for 7 days in control (+Fe+P) media, and then transferred to +Fe+P, Fe+P, or −Fe−P. Shoot tissues were collected at 76 h after the transfer and then used for total RNA extraction as described in ref. [57]. Each experiment was conducted with 16 plants, and 4–6 plants were pooled for RNA extraction, resulting in 3–4 biological replicates. Two micrograms of the total RNA were used for reverse transcription (Promega) to synthesize cDNA using oligo(dT) primer (Promega). Real-time quantitative reverse-transcription PCR (qRT-PCR) was performed as described in ref. [57] using 384-well plates with a LightCycler 480 Real-Time PCR System (Roche diagnostics). The Ubiquitin 10 mRNA (UBQ10: At4g05320) was used as a control

to calculate the relative mRNA level of each gene. The primers used in this study are listed in Supplementary Table 1.

**Genome-wide association studies (GWAS).** Gene expression data of the 32 genes that were specifically downregulated by −Fe+P but not by −Fe−P relative to +Fe+P were downloaded from leaf expression data of 727 Arabidopsis accessions[17]. Normalized RNA-seq read counts of these genes were used to perform principal component analysis, and contributions of the accessions to PC1 that explained 89.5% of the expression variance of the 32 genes were used to run genome-wide association (GWA) analysis. GWA mapping was performed using 1001 genomes SNP data[58] as implemented in the web application GWAPP[59]. Bonferroni correction ($\alpha = 0.05$) and false-discovery rate (FDR) at 5%[60] were implemented to account for multiple hypothesis tests.

**In planta transactivation assay.** The in planta transactivation assay[61] was performed in Nicotiana benthamiana. Promoters of PHOTOSYSTEM II SUBUNIT T (PSBTN); PHOTOSYSTEM II 5 kD, PLASTID TRANSCRIPTIONALLY ACTIVE 16 (PTAC16), LIGHT HARVESTING COMPLEX PHOTOSYSTEM II (LHCB4.1), PHOTOSYSTEM II REACTION CENTER W (PSBW) were fused to the reporter gene encoding β-glucuronidase (GUS) using Gateway cloning. The coding sequence of bZIP58 was inserted downstream of the CaMV 35S promoter (35S::bZIP58). The 35S::C-YFP construct was used as a negative effector control[61]. Each construct was transformed into A. tumefaciens. Cultures inoculated with positive clones were grown overnight at 28 °C, pelleted through centrifugation, and washed four times with infiltration buffer (10 mM $MgCl_2$, 10 mM MES (pH 5.6), and 100 µM acetosyringone). Fully expanded leaves of 5- to 6-week-old tobacco plants were co-infiltrated with the effector construct and reporter construct ($OD_{600}$ of 0.8). One leaf per plant was infiltrated for each plasmid combination and three independent infiltrations per combination were performed. Leaves were collected 3 days after infiltration and the infiltrated areas in each leaf were excised and pooled into one sample. To measure GUS enzymatic activity, 100 µg of total protein extract was incubated at 37 °C in GUS assay solution (2 mM 4-methylumbelliferyl β-D-glucuronide (Gold Biotechnology); 100 µl of the reaction was transferred to 1.9 ml of 0.2 M carbonate ($Na_2CO_3$) stop solution at the 0, 30, 60, 90, and 120 time points). GUS activity was measured using the Dyna Quant 200 fluorometer (Hoefer) and activity values were calculated following the β-glucuronidase (GUS) Fluorescent Activity Detection Kit (Sigma-Aldrich) protocols. For each promoter, the effect of bZIP58 on GUS activity was normalized to the effect of C-YFP on GUS activity to obtain a relative GUS enzymatic activity measurement. The primers used to clone the promoters of the photosynthesis-related genes are listed in Supplementary Table 1.

**Statistical analysis.** Box plots were generated using a web-based application "BoxPlotR"[62]. Statistical analyses of the data were performed using analysis of variance (ANOVA). One-way ANOVA with a Duncan post hoc test and two-way analysis of variance (ANOVA) and Tukey's honest significant difference (HSD) test were used to compare mean values. For all the statistical analyses, the difference was considered statistically significant when the test yielded a $P$ value <0.05.

**Reporting summary.** Further information on research design is available in the Nature Research Reporting Summary linked to this article.

## Data availability

Data supporting the findings of this work are available within the paper and its Supplementary Information and Source Data files. The datasets and plant materials generated and analyzed during this study are available from the corresponding author H.R. upon request. Transcriptome data were deposited in NCBI's Gene Expression Omnibus (GEO) under the project number GSE163190. Source data are provided with this paper.

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

## Acknowledgements

The authors thank the members of the Rhee lab; Benoit Lacombe and HONUDE team (INRAe) for their comments on the manuscript and helpful discussions. The authors are grateful to Jean-Francois Briat, Stéphane Mari and Diego Melo-Almeida for helpful discussion on Fe staining in Arabidopsis leaves. We thank the ICP-MS/TIMS Facility within Stanford University for assistance with the ICP-MS measurements, and the Stanford Functional Genomics Facility for assistance with RNA sequencing (Stanford, CA). The GWAS analysis was made possible using expression data generated by Kawakatsu et al.[17]. This work was

funded in part by the "Institut National de la Recherche Agronomique—Montpellier—France" INRA, the AgreeenSkills Plus, and Michigan State University (USA) to H.R. as well as by the Carnegie Institution for Science, Brigitte Berthelemot, National Science Foundation (IOS-1546838, IOS-1026003), and the U.S. Department of Energy, Office of Science, Office of Biological and Environmental Research, Genomic Science Program grant nos. DE-SC0018277, DE-SC0008769, and DE-SC0020366 to S.Y.R. The funders had no role in study design, data collection and analysis, decision to publish, or preparation of the manuscript.

## Author contributions

S.Y.R. and H.R. conceived the project. Experiments were designed by S.Y.R., H.R., H.N., and Z.S. and mainly carried out by H.N. S.C. performed and analyzed experiments related to photosystem II activity. bZIP58-GFP localization, ascorbic acid quantification, and hydrogen peroxide assays were conducted by H.C. and N.B. RNA-seq data were generated and analyzed by Y.D. Gene Ontology analysis was performed by K.Z. Z.S. performed the genome-wide association mapping. H.R. and K. L.P. performed the qRT-PCR analyses, generated plasmid constructs, the homozygote mutants, and the complemented mutant lines. S.Y.R., H.R., and Z.S. wrote the paper with input from all authors.

## Competing interests

The authors declare no competing interests.
