## [Peer Review File · Nature Communications]

Interdependent Iron and Phosphorus Availability Controls Photosynthesis Through Retrograde SignalingREVIEWER COMMENTS

Reviewer #1 (Remarks to the Author):

In their manuscript, Nam et al. address the long-standing question as to how iron and phosphate signaling pathways are connected. The authors report that omitting phosphate from the media results in a substantial decrease of chlorosis symptoms of iron-deficient plants and allow plants to maintain photosynthetic activity in the absence of iron. Using a combination of transcriptional profiling and GWAS, the authors identify three genes, the putative ascorbate transporter PHT4;4, the ascorbate biosynthesis gene VT,C and the transcription factor bZIP58 as being causative for their observations, and suggest a (not clearly defined) retrograde signaling cascade as an underlying mechanism for the observed 'stay-green' phenotype under iron- and phosphate-deficient conditions.

While the findings are interesting and novel and the work is well-conducted and described, there are additional or alternative interpretations of the results that should be addressed experimentally and discussed in the manuscript.

1) The authors state (line 82-83) that "the lack of chlorosis under -Fe-P is unlikely caused by more Fe available in shoots", a statement based on the lack of pronounced differences in Fe concentrations between the -Fe+P and -Fe-P regimes. This interpretation was supported by a lower FER1 expression in the latter treatment, leading to the conclusion (line 95) 'that the stay-green phenotype under the combined -Fe-P deficiency cannot be linked to Fe nutritional status in leaves.' This assumption is not sufficiently justified by the data. Firstly, a lack of correlation between chlorosis symptoms and iron content in leaves has been repeatedly described as 'chlorosis paradox' and observed under diverse conditions among different species. Secondly, ascorbate is a strong reductant for ferric iron (Grillet et al., 2014 JBC), which would make the oxidation state of iron the decisive mechanism, a supposition which is supported by the possible formation of sparingly soluble iron-phosphate compounds in the presence of P in the media.

2) It should also be discussed that the -Fe-P regime is the more likely scenario to be met under natural condition; iron deficiency is often coupled to phosphate deficiency, for example in plants grown on alkaline soils. Thus, the repressive effect of P on the expression of bZIP58 is the observation that needs to be explained, rather than the fact that the -Fe-P regime is NOT repressing the transcription of the gene.

3) It is surprising that the strong (more than 10-fold) repression of bZIP58 expression is not mirrored in the diverse transcriptomes of roots from iron-deficient Arabidopsis plants. It must therefore be stated that the observation of the authors does not appear to be robust but due to some specific conditions in their experimental set up. This discrepancy between the finding of the authors and publicly available data should be addressed.

4) The expression of bZIP58 was shown to be strongly dependent on the media pH (Lager et al., 2010, PCE). It is thus possible that the differences in bZIP58 expression between the growth types is due to alterations in pH. Was the media pH monitored and sufficiently during the experiment? Since alterations of the apoplastic pH have been reported for both iron- and phosphate-deficient plants, the influence on pH on the expression of bZIP58 should be experimentally addressed.

Reviewer #2 (Remarks to the Author):

Iron deficiency-mediated ROS production causes chlorosis. The authors showed that iron-phosphorus combined deficiency induces the production and chloroplastic transport of ascorbate via bZIP58, leading to prevention of chlorosis. The proposed retrograde signaling mechanism is interesting.

Major comments:

1) The authors should show the change of gene expression such as bZIP58, VTC4, photosynthesis-related genes in a time-dependent manner. Does bZIP58 expression increase at the beginning of -Fe-P condition?

2) PHT4;4 transports not only ascorbate but also Pi. Moreover, Fe + Pi deficiency increase bZIP58 and PHT4;4 expression. Is this effect directly associated with PHT4;4-mediated Pi transport?

3) How did other chlorophyll fluorescence parameters, such as non-photochemical quenching, change under several Fe and Pi conditions?

4) Is Fe and Pi deficiency-mediated retrograde signaling associated with genome uncoupled (gun) gene?

Minor comments:

1) In supplemental Figure 1, the authors mistook typing. [0,--] and [0.--].

Reviewer #3 (Remarks to the Author):

Review of the manuscript entitled "Interdependent Iron and Phosphorus Availability Controls Photosynthesis Through Retrograde Signaling"

In a previous study, involving some of the authors, the effect of single and multiple nutrient stress due to limited availability of Fe, Pi and Zn was examined. The authors showed that the negative effect of Fe starvation was alleviated by concomitant Pi depletion (or Pi and Zn depletion) in rice and highlighted a possible role of the Pi transporter OsPHO1;1 gene in coordinating pathways involved in Fe transport and Pi (and Zn) signaling (Saenchai et al. 2016. *Plant Sci.* 7:396. doi: 10.3389/fpls.2016.00396). In the present study, the authors focus on the mechanism that underlies the chloroplast response to concomitant Fe and (inorganic) P depletion leading to the 'stay-green' phenotype. Under -Fe-P condition, chlorophyll content was similar to +Fe+P control in different plants, but Fe content and ferritin 1 gene (AtFER1) activity measured on Arabidopsis shoots were low in Fe deficiency both with and without P. Chlorophyll content and PSII function decline were monitored in time course experiments, and three time points better examined by means of RNA sequencing. Then, the study was focused on a sub-set of genes specifically downregulated in -Fe+P and not in -Fe-P, almost all related to chloroplast function, and thus suggesting a relationship to chlorotic phenotype observed only in -Fe+P condition. The analysis of their expression in a worldwide collection of Arabidopsis accessions showed a positively correlation each other. Genome-wide association study (GWAS) was performed to identify 38 QTLs, and in particular the gene PHT4;4, encoding for an ascorbate (ASA) transporter localized at the chloroplast envelop membrane. PHT4;4 has a role in the xanthophyll cycle during photosynthesis and shows enhanced gene expression upon photostress. Experiments with a null *pht4;4* allele mutant and the complemented line, and with VTC4 mutant, which reduces the ASA level in leaves, indicated that ASA biosynthesis and transport into the chloroplast is important for the 'stay green' phenotype in Fe deficiency. On the other hand, VTC4 was downregulated in Fe deficiency when P was present, and ASA addition prevented the downregulation of photosynthesis genes. GWAS analysis identified a second gene, encoding for the transcription factor bZIP58, which was analyzed by qPCR, mutant and complemented lines, GFP-chimera localization. The photosynthesis genes, as well as ASA biosynthesis mediated by VTC4, appeared to be controlled by bZIP58, which was downregulated in Arabidopsis shoot under -Fe+P. The authors concluded proposing a model where Fe deficiency causes a decrease of ASA content in the chloroplast in the presence of P, with the consequence of a chlorotic phenotype. ROS should accumulate in this condition, with a detrimental effect on bZIP58 itself and, in turn, the photosynthesis genes.

Fe accumulates in the chloroplast playing a pivotal role as cofactor in photosystem complexes and almost all the component of the electron transport chain, despite this the most attention has been paid in the past on Fe deficiency effect at root level. Moreover, plants must adapt to a fluctuating availability of nutrients, and the identification of key genes involved in the co-regulation of micro- and macronutrient homeostasis is of major interest. Thus, the outcomes of this work are interesting, and the conclusions supported. Moreover, a lot of work using high standard methodologies has been done.

However, the major issue concerning the actual Fe availability in the presence of P and/or the effect of P when Fe is scarce remains to be discussed. Detailed comments are following.

Line 78 and following, Figure 1E, F. It's clear that the focus is on -Fe-P condition, but the increase of Fe content and AtFER1 gene expression under +Fe-P is significantly high, and authors should spend a few words to explain and/or support this finding with citation (metal accumulation in -P is normally observed? Which is the more accredited explanation to this? Which is the phenotype under this condition, as a high Fe content might generate an oxidative stress? etc). A high Fe content under P depletion raises the more general question if Fe availability is higher under P depletion than in P presence, and/or if Fe transport to leaves is higher. P is a macronutrient that could form with Fe scarcely soluble Fe-P compounds. In my opinion, this issue was not correctly addressed in the previous work (Saenchai et al. 2016) and, at least in part, might explain the lack of chlorosis observed also in this work for -Fe-P condition.

Line 80 and following. If I have understood, plants were grown in Fe-deficient media from the beginning, thus it is not surprising to find similar low Fe content in -Fe+P and -Fe-P. Ferritin are plastid-located proteins, and I'm wondering if it's correct to use FER1 expression as a read-out for 'free Fe in leaves'. As regards the link between Fe total content and FER1 expression, Bournier et al (cited as ref.12) showed that Fe distribution and ferritin gene expression could vary independently from total Fe content in the leaves. Even if AtFER1 expression is lower under -Fe-P compared to -Fe+P condition, this could reflect a lower content of 'free Fe' just in the chloroplast, while Fe content in the outside environment could be higher (in fact, total Fe content in leaves doesn't change in the two conditions). Moreover, we have no information about the actual bioavailability of leaf Fe in the two conditions, and it should be taken into account to explain the 'stay green' phenotype.

Line 121. 671 or 673 transcripts? See Venn diagram. 673 and 2434 transcripts: uniquely or specifically differentially regulated?

Line 126 and following. It is not clear to me where the 52 genes come from. Is this a sub-set of downregulated -Fe-P genes among 132 (52h) and 759 (76h) genes, and common to both time points? Same consideration for following sentences.

Line 151 and above. GWAS methodology is not in my area of expertise, and is not clear to me the passage from the subset of 32 genes, the accessions and the identification of 38 QTLs, among which the PHT4;4 containing QTL.

Line 166-203. Results show that transport of ASA into the chloroplast is important in Fe deficiency to recover the 'stay green' phenotype also when P is present (Fig. 3B). Fe is a cofactor of the photosystem complexes and almost all the component of the electron transport chain in the thylakoids, it is likely the

activation under light of ROS scavenging systems in Fe deficiency. Thus, what is not clear at all is why in -Fe+P plants the biosynthesis of ascorbate is downregulated and the ascorbate content lower (Figure 3D, E).

Line 172. Authors should indicate the gene ID for VTCs genes in the text, or at least in the legend and/or M&M.

Line 226. The expression value of AT1G13600 in Figure 4A is not consistent with the strong downregulation shown in 4B.

Line 241. ASA addition doesn't rescue the chlorotic phenotype in absence of bZIP58. Besides the control on VTC4, bZIP58 might affect ASA transport into the chloroplast. As a consequence, its downregulation in -Fe+P should result in a low ASA content and the impairment of ROS scavenging system. If authors agree, the model has to be changed indicating a lower ASA level in -Fe -P plants.

Line 410. Categories represented in the list are not only biological processes.

Line 494 and 685 (M&M), Figure S2A-F. It seems to me that the P value cutoff in the plot and Fold Change value do not correspond to what indicated in the legend and in M&M.

Reviewer #4 (Remarks to the Author):

In this study, authors have discovered that the interdependent effects of Fe and P availability on chlorosis are conserved across monocot and eudicot species, and used RNA-seq experiments and eGWAS to identify two candidate genes (PHT4;4 and bZIP58), which prevent the downregulation of photosynthesis genes leading to the stay-green phenotype under iron-phosphorus deficiency. These findings are very interesting and will help to understand the adaptability of photosynthesis to nutrient utilization. However, there are still a few comments that the authors should address.

Major comments:

1. For the GWAS results, theoretically, any gene within the QTL range may be a potentially functional gene. Authors should explain the basis of screening candidate genes clearly.

2. Although the regulation of bZIP58 on photosynthesis-related genes was illustrated by phenotypes and transcriptome data of mutants under different nutritional conditions, I suggest that additional molecular experiments, such as protein interaction experiments, are needed to further verify the regulation of photosynthesis-related genes.

Other comments:

1. Figure 2A, the unit of chlorophyll content on the ordinate is incorrectly marked, the author needs to check carefully.

2. Figure 3A, “-log(P-value)” should be changed to “-log₁₀(P-value)”.

3. Figure 4H, the unit of H₂O₂ content on the ordinate is incorrectly marked, the author needs to check carefully.

Dear Editor and Reviewers,

We are grateful to all reviewers for their constructive comments. We have now carried out a series of additional experiments to address their comments fully, which we believe have substantially improved the readability and impact of the manuscript. Most importantly, we have made new transgenic lines expressing bZIP58 under the control of a ROS-inducible promoter (*JUNGBRUNNEN1*; *JUB1*) to demonstrate ROS as a retrograde signal to inhibit photosynthesis-related genes and cause chlorosis in iron deficient environments (Figure 5). These experiments took a considerable amount of time, but we are very glad that we performed them because they strengthened our results and conclusions and raised the work's potential impact to another level.

The original manuscript was in a very short generic format. We have now expanded the text according to the Nature Communications format and added new figures. This allowed us to clarify and better explain some of the topics raised by the reviewers. We hope that the revised version satisfies any concerns raised.

Reviewer 1

In their manuscript, Nam et al. address the long-standing question as to how iron and phosphate signaling pathways are connected. The authors report that omitting phosphate from the media results in a substantial decrease of chlorosis symptoms of iron-deficient plants and allow plants to maintain photosynthetic activity in the absence of iron. Using a combination of transcriptional profiling and GWAS, the authors identify three genes, the putative ascorbate transporter PHT4;4, the ascorbate biosynthesis gene VT,C and the transcription factor bZIP58 as being causative for their observations, and suggest a (not clearly defined) retrograde signaling cascade as an underlying mechanism for the observed 'stay-green' phenotype under iron- and phosphate-deficient conditions.

While the findings are interesting and novel and the work is well-conducted and described, there are additional or alternative interpretations of the results that should be addressed experimentally and discussed in the manuscript.

1) The authors state (line 82-83) that "the lack of chlorosis under -Fe-P is unlikely caused by more Fe available in shoots", a statement based on the lack of pronounced differences in Fe concentrations between the -Fe+P and -Fe-P regimes. This interpretation was supported by a lower FER1 expression in the latter treatment, leading to the conclusion (line 95) 'that the stay-green phenotype under the combined -Fe-P deficiency cannot be linked to Fe nutritional status in leaves.' This assumption is not sufficiently justified by the data. Firstly, a lack of correlation between chlorosis symptoms and iron content in leaves has been repeatedly described as 'chlorosis paradox' and observed under diverse conditions among

different species. Secondly, ascorbate is a strong reductant for ferric iron (Grillet et al., 2014 JBC), which would make the oxidation state of iron the decisive mechanism, a supposition which is supported by the possible formation of sparingly soluble iron-phosphate compounds in the presence of P in the media.

Response: We now provide new evidence supporting our conclusion that the lack of chlorosis under -Fe-P is not due to higher bioavailable Fe in shoots.

To examine whether the 'stay green' phenotype in -Fe-P is linked to higher accumulation of bioavailable Fe (Fe^{2+}), we employed two complementary methods to quantify Fe^{2+} in plants. First, we stained for Fe^{2+} in leaves using Turnbull/DAB. Second, we quantified Fe^{2+} using the phenanthroline method. We did not find any difference in bioavailable Fe^{2+} accumulation under -Fe+P and -Fe-P conditions, strengthening our conclusion that the "stay green" phenotype under combined Fe and P deficiency is not linked to Fe nutritional status of leaves (Fig. 1).

Changes: Main Text: Page 3, Lines 73-79; Figure 1, Figure Legend: Page 23, Lines 743-747.

2) It should also be discussed that the -Fe-P regime is the more likely scenario to be met under natural condition; iron deficiency is often coupled to phosphate deficiency, for example in plants grown on alkaline soils. Thus, the repressive effect of P on the expression of *bZIP58* is the observation that need to be explained, rather than the fact that the -Fe-P regime is NOT repressing the transcription of the gene.

Response: To examine whether P availability regulates *bZIP58* expression, we profiled the expression of this gene under singular P deficiency. Control medium contained 1mM KH_2PO_4 as the sole source of P and in the P deficient medium, KH_2PO_4 was omitted. *bZIP58* was expressed similarly in +Fe+P and +Fe-P conditions (Supplementary Fig. 10B). This result is consistent with all publicly available transcriptome data of Arabidopsis grown under -P conditions (Misson et al., PNAS 2005; Morcuende et al., PCE 2007; Müller et al., Plant Physiol 2007) including our previous work published in 2011 (Rouached et al., Plant Journal 2011). Therefore, we conclude that P availability does not regulate *bZIP58* expression. In our original submission, we showed: 1) -P prevents *bZIP58* downregulation by -Fe; 2) exogenous application of H_2O_2 causes downregulation of *bZIP58* expression under the combined deficiency of Fe and P (-Fe-P); and 3) ascorbic acid (AsA) supplementation prevents the downregulation of *bZIP58* under -Fe+P. Therefore, we tested how the supply of H_2O_2 or AsA influences *bZIP58* expression under varying P availability. We found that P availability does not affect how H_2O_2 and AsA modulate *bZIP58* expression (Supplementary Fig. 10).

Changes: Main Text: Page 7, Lines 215-217. Supplementary Information: Supplementary Figure 10B, Figure Legend: Page 13, Lines 95-98.

3) It is surprising that the strong (more than 10-fold) repression of *bZIP58* expression is not mirrored in the diverse transcriptomes of roots from iron-deficient Arabidopsis

plants. It must therefore be stated that the observation of the authors does not appear to be robust but due to some specific conditions in their experimental set up. This discrepancy between the finding of the authors and publicly available data should be addressed.

Response: Our results show that *bZIP58* expression is significantly downregulated by Fe limitations in « shoots » of 7-day-old plants transferred to -Fe for 39, 52, and 76h. This finding is consistent with the limited number of available transcriptome data from « leaves » of Fe-deficient plants. For instance, we searched the data from Rodríguez-Celma et al., *Frontiers in Plant Sciences* 2013, and Grillet et al. *Nature Plants* 2018, and found that *bZIP58* is downregulated (≈ 2 fold) in -Fe conditions. We recognize that the downregulation of *bZIP58* in these shoot transcriptomes is less pronounced than in our study, but this could be due to differences in the experimental setup. Nevertheless, published data agree with our findings of consistent downregulation of *bZIP58* by Fe deficiency in shoots.

To address the Reviewer's comment, we analyzed *bZIP58* expression in « roots » of Fe-deficient plants in our growth conditions (Supplementary Fig. 10C). In comparison to shoots, we found a much weaker downregulation of *bZIP58* by -Fe in roots which may explain why *bZIP58* does not appear as one of the most strongly downregulated genes in transcriptomes of roots from iron-deficient *Arabidopsis* plants.

Changes: Main Text: Page 7, Lines 217-218. Supplementary Information: Supplementary Figure 10C, Figure Legend: Page 13, Lines 98-102.

4) The expression of *bZIP58* was shown to be strongly dependent on the media pH (Lager et al., 2010, PCE). It is thus possible that the differences in *bZIP58* expression between the growth types is due to alterations in pH. Was the media pH monitored and sufficiently during the experiment? Since alterations of the apoplastic pH have been reported for both iron- and phosphate-deficient plants, the influence on pH on the expression of *bZIP58* should be experimentally addressed.

Response: Yes, we adjusted pH of growth media used in this study to 5.5. We agree that Fe or P deficiency can influence rhizospheric pH, and a change in pH may regulate *bZIP58* expression in *Arabidopsis*. To clarify the situation, we profiled the expression of *bZIP58* in shoots and roots of plants grown in different pH. We found that *bZIP58* expression in roots is lower at pH 7 than at pH 5.5. However, we did not find any change in *bZIP58* expression in shoots in response to pH variation (Response Figure 1). Therefore, pH changes that might have been caused by Fe deficiency cannot account for the downregulation of *bZIP58* in shoots.

Response Figure 1. Relative mRNA abundance of bZIP58 in roots (A) and shoots (B) of 7-day-old wild-type plants grown on +Fe+P media adjusted to pH 5.5 or 7.0.

For now, we do not see the relevance of these data with our findings, therefore we did not to include these new results in the paper.

Reviewer 2

Iron deficiency-mediated ROS production causes chlorosis. The authors showed that iron-phosphorus combined deficiency induces the production and chloroplastic transport of ascorbate via bZIP58, leading to prevention of chlorosis. The proposed retrograde signaling mechanism is interesting.

Major comments:

1) The authors should show the change of gene expression such as bZIP58, VTC4, photosynthesis-related genes in a time-dependent manner. Does bZIP58 expression increase at the beginning of -Fe-P condition?

Response: The expression data of the 32 photosynthesis-related genes (PRGs) across different treatments at different time points (39h, 52h, 76h) are presented as a heatmap in Figure 2E. RNA-seq data have been deposited to the NCBI's Gene Expression Omnibus (GEO). Furthermore, we profiled the expression of VTC4 and bZIP58 using qRT-PCR on plants transferred from the control (+Fe+P) condition to: +Fe+P, -Fe+P, or -Fe-P for 39h, 52h, or 76h (Supplementary Fig. 10A). While we find repression of bZIP58 by -Fe+P, no change in the expression of bZIP58 is detected at any of these time points under -Fe-P conditions. VTC4 expression was gradually induced in response to -Fe-P during these time points.

Changes: Main Text: Page 7, Lines 214-215. Supplementary Information: Supplementary Figure 10A, Figure Legend: Page 13, Lines 93-95.

2) PHT4;4 transports not only ascorbate but also Pi. Moreover, Fe + Pi deficiency increase bZIP58 and PHT4;4 expression. Is this effect directly associated with PHT4;4-mediated Pi transport?

Response: In our study, the expression of *bZIP58* was downregulated by -Fe+P (Fig. 4B). Our transcriptome data show that -Fe+P did not alter *PHT4;4* expression.

PHT4;4 is known to transports inorganic phosphate (Pi) as well as ascorbate (AsA) to the chloroplasts (Guo et al., New Phytol. 2008 ; Miyaji et al., Nature Comm 2015). To test whether perturbation in chloroplastic Pi transporters (*PHT4;3*, *PHT4;5*, and *PHT4;6*) (Guo et al., New Phytol. 2008) influences chlorophyll accumulation under -Fe-P, we characterized knock-out mutants of these genes for chlorophyll accumulation under different conditions. None of the Pi transporter mutants differed significantly from wild-type plants for chlorophyll content under -Fe-P, showing that perturbation of Pi transport to the chloroplast is not associated with the stay-green phenotype. Furthermore, plants lacking key genes that regulate Pi homeostasis, the transcription factor *PHOSPHATE RESPONSE 1 (PHR1)* and the *SPX DOMAIN GENE 1 (SPX1)*, exhibited similar chlorophyll content as wild-type plants in -Fe-P (Supplementary Fig. 9). Given that none of the Pi transport/homeostasis related genes affects chlorophyll accumulation in -Fe-P (Supplementary Fig. 9), *pht4;4* mutant phenotypes indicate that these effects are mediated through its AsA transport activity rather than Pi transport activity. Our conclusion is further supported by the prevention of chlorosis by AsA supplementation in -Fe+P in wild-type plants but not in *pht4;4* mutant plants.

Changes. Main Text: Page 5, Lines 155-164. Supplementary Information: Supplementary Figure 9, Figure Legend: Page 12, Lines 82-91.

3) How did other chlorophyll fluorescence parameters, such as non-photochemical quenching, change under several Fe and Pi conditions?

Response: We have analyzed how Fe and P availability influence non-photochemical quenching (NPQ). Plants under -Fe-P showed slightly lower but stabilized NPQ compared to those in +Fe+P. -Fe+P caused a strong and significant decrease in NPQ observable starting at 76h after the transfer to -Fe+P conditions (Supplementary Fig. 2D).

Changes. Main Text: Page 4, Line 98-100. Supplementary Information: Supplementary Figure 2D, Figure Legend: Page 5, Lines 19-20.

4) Is Fe and Pi deficiency-mediated retrograde signaling associated with genome uncoupled (*gun*) gene?

Response: Our transcriptome analysis did not show any changes in the expression of *GUN1* (AT2G31400) under -Fe+P or -Fe-P conditions. Previous work by Patrice A Salomé, Michele Oliva, Detlef Weigel, and Ute Krämer (EMBO J (2013) 32:511-523)

concluded that: "GUN1 does not play a general role in seedlings grown on various Fe supply". Therefore, it is unlikely that Fe and P deficiency-mediated retrograde signaling is associated with GUN1.

Minor comments:

1) In supplemental Figure 1, the authors mistook typing. [0,--] and [0.--].

Response: Corrected.

Reviewer 3

Review of the manuscript entitled "Interdependent Iron and Phosphorus Availability Controls Photosynthesis Through Retrograde Signaling"

In a previous study, involving some of the authors, the effect of single and multiple nutrient stress due to limited availability of Fe, Pi and Zn was examined. The authors showed that the negative effect of Fe starvation was alleviated by concomitant Pi depletion (or Pi and Zn depletion) in rice and highlighted a possible role of the Pi transporter OsPHO1;1 gene in coordinating pathways involved in Fe transport and Pi (and Zn) signaling (Saenchai et al. 2016. Plant Sci. 7:396. doi: 10.3389/fpls.2016.00396). In the present study, the authors focus on the mechanism that underlies the chloroplast response to concomitant Fe and (inorganic) P depletion leading to the 'stay-green' phenotype. Under -Fe-P condition, chlorophyll content was similar to +Fe+P control in different plants, but Fe content and ferritin 1 gene (AtFER1) activity measured on Arabidopsis shoots were low in Fe deficiency both with and without P. Chlorophyll content and PSII function decline were monitored in time course experiments, and three time points better examined by means of RNA sequencing. Then, the study was focused on a sub-set of genes specifically downregulated in -Fe+P and not in -Fe-P, almost all related to chloroplast function, and thus suggesting a relationship to chlorotic phenotype observed only in -Fe+P condition. The analysis of their expression in a worldwide collection of Arabidopsis accessions showed a positively correlation each other. Genome-wide association study (GWAS) was performed to identify 38 QTLs, and in particular the gene PHT4;4, encoding for an ascorbate (ASA) transporter localized at the chloroplast envelop membrane. PHT4;4 has a role in the xanthophyll cycle during photosynthesis and shows enhanced gene expression upon photostress. Experiments with a null pht4;4 allele mutant and the complemented line, and with VTC4 mutant, which reduces the ASA level in leaves, indicated that ASA biosynthesis and transport into the chloroplast is important for the 'stay green' phenotype in Fe deficiency. On the other hand, VTC4 was downregulated In Fe deficiency when P was present, and ASA addition prevented the downregulation of photosynthesis genes. GWAS analysis identified a second gene, encoding for the

transcription factor bZIP58, which was analyzed by qPCR, mutant and complemented lines, GFP-chimera localization.

The photosynthesis genes, as well as ASA biosynthesis mediated by VTC4, appeared to be controlled by bZIP58, which was downregulated in Arabidopsis shoot under -Fe+P. The authors concluded proposing a model where Fe deficiency causes a decrease of ASA content in the chloroplast in the presence of P, with the consequence of a chlorotic phenotype. ROS should accumulate in this condition, with a detrimental effect on bZIP58 itself and, in turn, the photosynthesis genes.

Fe accumulates in the chloroplast playing a pivotal role as cofactor in photosystem complexes and almost all the component of the electron transport chain, despite this the most attention has been paid in the past on Fe deficiency effect at root level. Moreover, plants must adapt to a fluctuating availability of nutrients, and the identification of key genes involved in the co-regulation of micro- and macronutrient homeostasis is of major interest. Thus, the outcomes of this work are interesting, and the conclusions supported.

Moreover, a lot of work using high standard methodologies has been done.

However, the major issue concerning the actual Fe availability in the presence of P and/or the effect of P when Fe is scarce remains to be discussed. Detailed comments are following.

Line 78 and following, Figure 1E, F. It's clear that the focus is on -Fe-P condition, but the increase of Fe content and AtFER1 gene expression under +Fe-P is significantly high, and authors should spend a few words to explain and/or support this finding with citation (metal accumulation in -P is normally observed? Which is the more accredited explanation to this? Which is the phenotype under this condition, as a high Fe content might generate an oxidative stress? etc).

Response: The accumulation of Fe under P deficiency has been consistently observed in several studies (i.e. Misson et al., PNAS 2005; Ward et al., Plant Physiology 2008; Saenchai et al., Front Plant Sci. 2016; Shi et al. Front Plant Sci. 2018). P deficiency causes a decrease in shoot biomass associated with accumulation of anthocyanin (Poirier and Buchner, The Arabidopsis Book: e0024, 2002). A pronounced phenotype is also observed at the root level: inhibition of the primary root elongation, and production of more lateral root and root hairs (Bouain et al., Current Genomics 2018). Remarkably, root growth of plants under simultaneous Fe and P deficiency is similar to root growth of plants grown under +Fe+P (Ward et al., Plant Physiology 2008, Balzergue et al., Nature Comm 2017). Müller and colleagues showed that the inhibition of root growth under -P was caused by high Fe accumulation which generates ROS (Müller et al., Developmental Cell 2015).

A high Fe content under P depletion raises the more general question if Fe availability is higher under P depletion than in P presence, and/or if Fe transport to leaves is higher. P is a macronutrient that could form with Fe scarcely soluble Fe-P compounds. In my opinion, this issue was not correctly addressed in the previous work (Saenchai et al. 2016) and, at least in part, might explain the lack of chlorosis observed also in this work for -Fe-P condition.

Response: We now provide evidence that the stay-green phenotype in -Fe-P is not due to higher accumulation of bioavailable iron in leaves (See response 1 to the Reviewer 1).

Changes: Main Text: Page 3, Lines 73-79; Figure 1, Figure Legend: Page 23, Lines 743-747.

Line 80 and following. If I have understood, plants were grown in Fe-deficient media from the beginning, thus it is not surprising to find similar low Fe content in -Fe+P and -Fe-P. Ferritin are plastid-located proteins, and I'm wondering if it's correct to use FER1 expression as a read-out for 'free Fe in leaves'. As regards the link between Fe total content and FER1 expression, Bournier et al (cited as ref.12) showed that Fe distribution and ferritin gene expression could vary independently from total Fe content in the leaves. Even if AtFER1 expression is lower under -Fe-P compared to -Fe+P condition, this could reflect a lower content of 'free Fe' just in the chloroplast, while Fe content in the outside environment could be higher (in fact, total Fe content in leaves doesn't change in the two conditions). Moreover, we have no information about the actual bioavailability of leaf Fe in the two conditions, and it should be taken into account to explain the 'stay green' phenotype.

Response: We agree with the Reviewer, and we have replaced the *AtFER1* expression data with new data for accumulation of bioavailable Fe in shoots using two methods. Our results showed that the lack of chlorosis under -Fe-P is not associated with more bioavailable Fe in shoots. Please see response 1 to Reviewer 1.

Changes: Main Text: Page 3, Lines 73-79; Figure 1, Figure Legend: Page 23, Lines 743-747.

Line 121. 671 or 673 transcripts? See Venn diagram. 673 and 2434 transcripts: uniquely or specifically differentially regulated?

Response: Thank you. Corrected. 673 transcripts. We used specifically differentially regulated.

Changes: Main Text: Page 4, Line 110. Supplementary Information: Supplementary Figure 4, Figure Legend: Page 7, Lines 34-40.

Line 126 and following. It is not clear to me where the 52 genes come from. Is this a sub-set of downregulated -Fe-P genes among 132 (52h) and 759 (76h) genes, and common to both time points? Same consideration for following sentences.

Response: The 52 downregulated genes in -Fe-P are common to both time points 52h and 76h. The sentence in Line 126 is revised as follows: “The common set of 52 genes that were specifically downregulated by -Fe-P both at 52h and 76h after the transfer (Supplementary Fig. 5A, Supplementary Data) showed an enrichment for ribosomal genes (Supplementary Fig. 5B) while upregulated genes (162 genes, Supplementary Fig. 5C, Supplementary Data) revealed an enrichment for genes involved in cation transport, response to water and ester hydrolysis (Supplementary Fig. 5D, Supplementary Data)”.

Changes: Main Text: Page 4, Lines 114-119. Supplementary Information: Supplementary Figure 5, Figure Legend: Page 8, Lines 43-51.

Line 151 and above. GWAS methodology is not in my area of expertise, and is not clear to me the passage from the subset of 32 genes, the accessions and the identification of 38 QTLs, among which the PHT4;4 containing QTL.

Response: The methodology is as follows:

Step 1. We identified 32 genes that were specifically downregulated by -Fe+P based on our transcriptome data at different time points.

Step 2. We retrieved expression data of the 32 genes in published transcriptome of 727 Arabidopsis accessions.

Step 3. We performed Principal Component Analysis (PCA) to reduce the dimensionality of these expression data.

Step 4. We retained PC1, which explained 89.5% of the variation in expression of these genes across the Arabidopsis accessions.

Step 5. We used the PC1 scores of the accessions to perform a genome-wide association mapping.

Step 6. Our GWA analysis detected 38 QTLs containing 145 candidate genes, based on a 20-kb window per QTL and using a 5% false discovery rate (FDR) threshold.

Step 7. Functional validation of candidate genes of a selected QTL led to the identification of the inorganic phosphate/ascorbate transporter *PHT4;4* (AT4G00370) as a causal gene.

We overhauled the Results section to clarify this.

Changes: Main Text: Page 5, Lines 134-164.

Line 166-203. Results show that transport of ASA into the chloroplast is important in Fe deficiency to recover the ‘stay green’ phenotype also when P is present (Fig. 3B). Fe is a cofactor of the photosystem complexes and almost all the component of the electron transport chain in the thylakoids, it is likely the activation under light of ROS scavenging systems in Fe deficiency. Thus, what is not clear at all is why in -Fe+P

plants the biosynthesis of ascorbate is downregulated and the ascorbate content lower (Figure 3D, E).

Response: This is a very interesting point. We believe, the difference is due to the focus of our study on initial response to Fe deficiency stress in comparison to long-term adaptation. AsA levels have been shown to increase in response to 2 to 3 weeks Fe-deficiency treatments in *Brassica napus* leaves and sugar beet roots (Tewari et al, Environ. Expt. Bot., 2013; Zaharieva and Abadía, Protoplasma, 2003). However, 3 to 7 days Fe-deficiency treatment in *Arabidopsis* exhibited a slight reduction in AsA accumulation in shoots (Ramírez et al, J. Expt. Bot., 2013). In our study, we measured AsA content 52 hours after Fe deficiency treatment, and we found a significant decrease in AsA accumulation in leaves. It's intriguing to find a consistent reduction of AsA during the early phase of Fe-deficiency. One possibility is that AsA reduction could be one of the earliest signs of Fe deficiency stress. Supporting this hypothesis is a recent finding that an early response to salt stress is repression of AsA biosynthesis by ABI4-mediated repression of VTC2 (Kakan et al, BMC Plant Biology, 2021). Identifying the upstream factors of the signaling pathway we discovered, including AsA biosynthesis, will be an important future direction. We have discussed this point in the Discussion section of the revised manuscript.

Changes: Main Text: Page 11, Lines 319-328.

Line 172. Authors should indicate the gene ID for VTCs genes in the text, or at least in the legend and/or M&M.

Response: The gene IDs for VTCs genes are now indicated in the main text and Figure 3 legend.

Changes: Main Text: Page 6, Lines 168-169. Figure 3, Figure Legend: Page 24, Lines 783-784.

Line 226. The expression value of AT1G13600 in Figure 4A is not consistent with the strong downregulation shown in 4B.

Response: We now present data for 76h in 4A to be consistent with data shown in 4B.

Changes: Main Text: Figure 4A, page 30, Figure Legend: Page 24, Line 797.

Line 241. ASA addition doesn't rescue the chlorotic phenotype in absence of bZIP58. Besides the control on VTC4, bZIP58 might affect ASA transport into the chloroplast. As a consequence, its downregulation in -Fe+P should result in a low ASA content and the impairment of ROS scavenging system. If authors agree, the model has to be changed indicating a lower ASA level in -Fe -P plants.

Response: We now indicate a lower AsA level in -Fe-P plants in the figure. Furthermore, to provide definitive evidence for the presence of a plastidic signal,

ROS, which influences the expression of PRGs and chlorophyll content under -Fe+P by repressing bZIP58, we now expressed bZIP58 under the control of ROS-inducible promoter (*JUNGBRUNNEN1*; *JUB1*) to effectively create an antagonist version of *bZIP58*. The endogenous *bZIP58* in Col-0 is repressed by -Fe+P. However, -Fe+P upregulates *JUB1p::bZIP58* by 2.5-fold in *bzip58* null mutant, demonstrating the sensitivity of the *JUB1* promoter to -Fe+P-induced ROS accumulation (Fig. 5). Notably, a greatly enhanced (4-fold) induction of *JUB1p::bZIP58* is detected in *bzip58xph4;4* double mutant plants in -Fe+P, suggesting a greater chloroplastic ROS signal in the absence of the AsA transporter PHT4;4. Enhanced induction of *JUB1p::bZIP58* by -Fe+P in *bzip58xph4;4* compared to *bzip58* lines indicates that PHT4;4 acts downstream -Fe to influence the expression of the *JUB1p::bZIP58* in the nucleus. Consistently, AsA supplementation in -Fe+P prevents the induction of *JUB1p::bZIP58* in the *bzip58* mutant but not in the *bzip58 ph4;4* double mutant. Again, these results demonstrate that PHT4;4-mediated AsA transport to the chloroplast alters chloroplastic ROS that modulates the expression of *bZIP58* detected by the ROS-inducible promoter system. Finally, we show that under -Fe+P, the expression of *JUB1p::bZIP58* enhances chlorophyll accumulation and expression of PRGs in the *bzip58* mutant. These responses are exacerbated in the *bzip58xph4;4* double mutant expressing *JUB1p::bZIP58* (Fig. 5). These new results demonstrate that changes in chloroplastic ROS in -Fe+P are linked to chlorosis and expression of PRG, which is mediated by a nuclear transcription factor bZIP58.

Changes: Main Text: Figure 6, Page 32, Page 8, Lines 260-278; Figure 5. Figure Legend: Page 25, Lines 818-830.

Line 410. Categories represented in the list are not only biological processes.

Response: Thank you for pointing this out, which is correct. We included enrichments in all three branches of the GO. We have now removed the phrase “biological processes (GO-BP)”.

Changes: Main Text: Figure 2, Figure Legend: Page 23, Line 767.

Line 494 and 685 (M&M), Figure S2A-F. It seems to me that the P value cutoff in the plot and Fold Change value do not correspond to what indicated in the legend and in M&M.

Response: We have now corrected this. The axes are in log scales. Red: $\log_2\text{FoldChange} > 1$ and $-\log_{10}P > 6$; Blue: $\log_2\text{FoldChange} < 1$ and $-\log_{10}P > 6$; Green: $\log_2\text{FoldChange} > 1$ and $-\log_{10}P < 6$; Grey: $\log_2\text{FoldChange} < 1$ and $-\log_{10}P < 6$. A default cut-off of $\log_2\text{FoldChange} > 1$ and adjusted p-value $< 10^{-6}$ was used.

Changes: Supplementary Information: Supplemental Figure 3, Figure Legend: Page 6, Lines 30-33.

Reviewer #4 (Remarks to the Author):

In this study, authors have discovered that the interdependent effects of Fe and P availability on chlorosis are conserved across monocot and eudicot species, and used RNA-seq experiments and eGWAS to identify two candidate genes (*PHT4;4* and *bZIP58*), which prevent the downregulation of photosynthesis genes leading to the stay-green phenotype under iron-phosphorus deficiency. These findings are very interesting and will help to understand the adaptability of photosynthesis to nutrient utilization. However, there are still a few comments that the authors should address.

Major comments:

1. For the GWAS results, theoretically, any gene within the QTL range may be a potentially functional gene. Authors should explain the basis of screening candidate genes clearly.

Response: In our GWAS analysis, we followed up on the 2 most significant QTLs using a 5% false discovery rate (FDR) threshold. Within these two QTL regions, we examined all the genes in a 20-kb window and stopped when we found phenotypes of *pht4;4* and *bzip58* mutants. Precisely, the first QTL located in chromosome 1 (associated with SNP4653399) contains 9-candidate genes (AT1G13570 to AT1G13610) (Supplementary Fig. 7), and the second QTL located on the chromosome 4 (associated with SNP171674) contains 6 candidate genes (AT4G00355 to AT4G00400) (Supplementary Fig. 7). We systemically tested mutants of all the genes in these QTLs. Out of the 15 candidate genes we examined, knock-out mutants of *bZIP58* and *PHT4;4* exhibited phenotypes under different conditions. We have rewritten this part in Results section to clarify this point.

Change : Main Text : Page 5, Lines 143-149.

2. Although the regulation of *bZIP58* on photosynthesis-related genes was illustrated by phenotypes and transcriptome data of mutants under different nutritional conditions, I suggest that additional molecular experiments, such as protein interaction experiments, are needed to further verify the regulation of photosynthesis-related genes.

Response: To test whether *bZIP58* could “directly” regulate PRGs, we used an established method, quantitative *in planta* transactivation assay, which we routinely employ in our lab to identify target of transcription factors (Bossi et al., BMC Genomics (2017) 18:480). Briefly, selected PRG promoters were fused to a reporter gene (a β -glucuronidase (GUS)) and co-expressed in tobacco leaves with the *bZIP58* TF driven by a 35S promoter. 35S-YFP was used as a negative control. We studied five PRGs (*PHOTOSYSTEM II SUBUNIT T (PSBTN)*; *PHOTOSYSTEM II 5 kD*, *PLASTID TRANSCRIPTIONALLY ACTIVE 16 (PTAC16)*, *LIGHT HARVESTING COMPLEX PHOTOSYSTEM II (LHCB4.1)*, *PHOTOSYSTEM II REACTION CENTER*

W (PSBW)) during this analysis. We found that bZIP58 activates the promoter activity of 3 of the 5 genes tested (PSBTN, PHOTOSYSTEM II 5 kD and PTAC16) (Supplementary Fig. 11B). Therefore, bZIP58 can directly activate the expression of some PRGs. These results demonstrate that bZIP58 is a key gene to regulate the expression of PRGs in an Fe-dependent manner. We believe that figuring out how bZIP58 regulates “directly or indirectly” the PRGs is another paper.

Change : Main Text : Page 7, Lines 225-234. Supplementary Information: Supplementary Figure 11, Figure Legend: Page 14, Lines 103-114.

Other comments:

1. Figure 2A, the unit of chlorophyll content on the ordinate is incorrectly marked, the author needs to check carefully.

Corrected.

2. Figure 3A, “-log(P-value)” should be changed to “-log₁₀(P-value)”.

Corrected.

3. Figure 4H, the unit of H₂O₂ content on the ordinate is incorrectly marked, the author needs to check carefully.

Corrected (now is Figure 4G).

Dear Editor and Reviewers,

We are grateful to all reviewers for their constructive comments. We have now carried out a series of additional experiments to address their comments fully, which we believe have substantially improved the readability and impact of the manuscript. Most importantly, we have made new transgenic lines expressing bZIP58 under the control of a ROS-inducible promoter (*JUNGBRUNNEN1*; *JUB1*) to demonstrate ROS as a retrograde signal to inhibit photosynthesis-related genes and cause chlorosis in iron deficient environments (Figure 5). These experiments took a considerable amount of time, but we are very glad that we performed them because they strengthened our results and conclusions and raised the work's potential impact to another level.

The original manuscript was in a very short generic format. We have now expanded the text according to the Nature Communications format and added new figures. This allowed us to clarify and better explain some of the topics raised by the reviewers. We hope that the revised version satisfies any concerns raised.

Reviewer 1

In their manuscript, Nam et al. address the long-standing question as to how iron and phosphate signaling pathways are connected. The authors report that omitting phosphate from the media results in a substantial decrease of chlorosis symptoms of iron-deficient plants and allow plants to maintain photosynthetic activity in the absence of iron. Using a combination of transcriptional profiling and GWAS, the authors identify three genes, the putative ascorbate transporter PHT4;4, the ascorbate biosynthesis gene VT,C and the transcription factor bZIP58 as being causative for their observations, and suggest a (not clearly defined) retrograde signaling cascade as an underlying mechanism for the observed 'stay-green' phenotype under iron- and phosphate-deficient conditions.

While the findings are interesting and novel and the work is well-conducted and described, there are additional or alternative interpretations of the results that should be addressed experimentally and discussed in the manuscript.

1) The authors state (line 82-83) that "the lack of chlorosis under -Fe-P is unlikely caused by more Fe available in shoots", a statement based on the lack of pronounced differences in Fe concentrations between the -Fe+P and -Fe-P regimes. This interpretation was supported by a lower FER1 expression in the latter treatment, leading to the conclusion (line 95) 'that the stay-green phenotype under the combined -Fe-P deficiency cannot be linked to Fe nutritional status in leaves.' This assumption is not sufficiently justified by the data. Firstly, a lack of correlation between chlorosis symptoms and iron content in leaves has been repeatedly described as 'chlorosis paradox' and observed under diverse conditions among

different species. Secondly, ascorbate is a strong reductant for ferric iron (Grillet et al., 2014 JBC), which would make the oxidation state of iron the decisive mechanism, a supposition which is supported by the possible formation of sparingly soluble iron-phosphate compounds in the presence of P in the media.

Response: We now provide new evidence supporting our conclusion that the lack of chlorosis under -Fe-P is not due to higher bioavailable Fe in shoots.

To examine whether the 'stay green' phenotype in -Fe-P is linked to higher accumulation of bioavailable Fe (Fe^{2+}), we employed two complementary methods to quantify Fe^{2+} in plants. First, we stained for Fe^{2+} in leaves using Turnbull/DAB. Second, we quantified Fe^{2+} using the phenanthroline method. We did not find any difference in bioavailable Fe^{2+} accumulation under -Fe+P and -Fe-P conditions, strengthening our conclusion that the "stay green" phenotype under combined Fe and P deficiency is not linked to Fe nutritional status of leaves (Fig. 1).

Changes: Main Text: Page 3, Lines 73-79; Figure 1, Figure Legend: Page 23, Lines 746-750.

2) It should also be discussed that the -Fe-P regime is the more likely scenario to be met under natural condition; iron deficiency is often coupled to phosphate deficiency, for example in plants grown on alkaline soils. Thus, the repressive effect of P on the expression of *bZIP58* is the observation that need to be explained, rather than the fact that the -Fe-P regime is NOT repressing the transcription of the gene.

Response: To examine whether P availability regulates *bZIP58* expression, we profiled the expression of this gene under singular P deficiency. Control medium contained 1mM KH_2PO_4 as the sole source of P and in the P deficient medium, KH_2PO_4 was omitted. *bZIP58* was expressed similarly in +Fe+P and +Fe-P conditions (Supplementary Fig. 10C). This result is consistent with all publicly available transcriptome data of Arabidopsis grown under -P conditions (Misson et al., PNAS 2005; Morcuende et al., PCE 2007; Müller et al., Plant Physiol 2007) including our previous work published in 2011 (Rouached et al., Plant Journal 2011). Therefore, we conclude that P availability does not regulate *bZIP58* expression. In our original submission, we showed: 1) -P prevents *bZIP58* downregulation by -Fe; 2) exogenous application of H_2O_2 causes downregulation of *bZIP58* expression under the combined deficiency of Fe and P (-Fe-P); and 3) ascorbic acid (AsA) supplementation prevents the downregulation of *bZIP58* under -Fe+P. Therefore, we tested how the supply of H_2O_2 or AsA influences *bZIP58* expression under varying P availability. We found that P availability does not affect how H_2O_2 and AsA modulate *bZIP58* expression (Supplementary Fig. 10).

Changes: Main Text: Page 7, Lines 216-218. Supplementary Information: Supplementary Figure 10C, Figure Legend: Page 13, Lines 100-103.

3) It is surprising that the strong (more than 10-fold) repression of *bZIP58* expression is not mirrored in the diverse transcriptomes of roots from iron-deficient Arabidopsis

plants. It must therefore be stated that the observation of the authors does not appear to be robust but due to some specific conditions in their experimental set up. This discrepancy between the finding of the authors and publicly available data should be addressed.

Response: Our results show that *bZIP58* expression is significantly downregulated by Fe limitations in « shoots » of 7-day-old plants transferred to -Fe for 39, 52, and 76h. This finding is consistent with the limited number of available transcriptome data from « leaves » of Fe-deficient plants. For instance, we searched the data from Rodríguez-Celma et al., *Frontiers in Plant Sciences* 2013, and Grillet et al. *Nature Plants* 2018, and found that *bZIP58* is downregulated (≈ 2 fold) in -Fe conditions. We recognize that the downregulation of *bZIP58* in these shoot transcriptomes is less pronounced than in our study, but this could be due to differences in the experimental setup. Nevertheless, published data agree with our findings of consistent downregulation of *bZIP58* by Fe deficiency in shoots.

To address the Reviewer's comment, we analyzed *bZIP58* expression in « roots » of Fe-deficient plants in our growth conditions (Supplementary Fig. 10D). In comparison to shoots, we found a much weaker downregulation of *bZIP58* by -Fe in roots which may explain why *bZIP58* does not appear as one of the most strongly downregulated genes in transcriptomes of roots from iron-deficient *Arabidopsis* plants.

Changes: Main Text: Page 7, Lines 218-219. Supplementary Information: Supplementary Figure 10D, Figure Legend: Page 13, Lines 103-107.

4) The expression of *bZIP58* was shown to be strongly dependent on the media pH (Lager et al., 2010, PCE). It is thus possible that the differences in *bZIP58* expression between the growth types is due to alterations in pH. Was the media pH monitored and sufficiently during the experiment? Since alterations of the apoplastic pH have been reported for both iron- and phosphate-deficient plants, the influence on pH on the expression of *bZIP58* should be experimentally addressed.

Response: Yes, we adjusted pH of growth media used in this study to 5.5. We agree that Fe or P deficiency can influence rhizospheric pH, and a change in pH may regulate *bZIP58* expression in *Arabidopsis*. To clarify the situation, we profiled the expression of *bZIP58* in shoots and roots of plants grown in different pH. We found that *bZIP58* expression in roots is lower at pH 7 than at pH 5.5. However, we did not find any change in *bZIP58* expression in shoots in response to pH variation (Response Figure 1). Therefore, pH changes that might have been caused by Fe deficiency cannot account for the downregulation of *bZIP58* in shoots.

Response Figure 1. Relative mRNA abundance of bZIP58 in roots (A) and shoots (B) of 7-day-old wild-type plants grown on +Fe+P media adjusted to pH 5.5 or 7.0.

For now, we do not see the relevance of these data with our findings, therefore we did not to include these new results in the paper.

Reviewer 2

Iron deficiency-mediated ROS production causes chlorosis. The authors showed that iron-phosphorus combined deficiency induces the production and chloroplastic transport of ascorbate via bZIP58, leading to prevention of chlorosis. The proposed retrograde signaling mechanism is interesting.

Major comments:

1) The authors should show the change of gene expression such as bZIP58, VTC4, photosynthesis-related genes in a time-dependent manner. Does bZIP58 expression increase at the beginning of -Fe-P condition?

Response: The expression data of the 32 photosynthesis-related genes (PRGs) across different treatments at different time points (39h, 52h, 76h) are presented as a heatmap in Figure 2E. RNA-seq data have been deposited to the NCBI's Gene Expression Omnibus (GEO). Furthermore, we profiled the expression of VTC4 and bZIP58 using qRT-PCR on plants transferred from the control (+Fe+P) condition to: +Fe+P, -Fe+P, or -Fe-P for 39h, 52h, or 76h (Supplementary Fig. 10A). While we find repression of bZIP58 by -Fe+P, no change in the expression of bZIP58 is detected at any of these time points under -Fe-P conditions. VTC4 expression was gradually induced in response to -Fe-P during these time points.

Changes: Main Text: Page 6, Lines 169-171; Page 7, Lines 213-216. Supplementary Information: Supplementary Figure 10A, Figure Legend: Page 13, Lines 93-96.

2) PHT4;4 transports not only ascorbate but also Pi. Moreover, Fe + Pi deficiency increase bZIP58 and PHT4;4 expression. Is this effect directly associated with PHT4;4-mediated Pi transport?

Response: In our study, the expression of *bZIP58* was downregulated by -Fe+P (Fig. 4B). Our transcriptome data show that -Fe+P did not alter *PHT4;4* expression.

PHT4;4 is known to transports inorganic phosphate (Pi) as well as ascorbate (AsA) to the chloroplasts (Guo et al., New Phytol. 2008 ; Miyaji et al., Nature Comm 2015). To test whether perturbation in chloroplastic Pi transporters (*PHT4;3*, *PHT4;5*, and *PHT4;6*) (Guo et al., New Phytol. 2008) influences chlorophyll accumulation under -Fe-P, we characterized knock-out mutants of these genes for chlorophyll accumulation under different conditions. None of the Pi transporter mutants differed significantly from wild-type plants for chlorophyll content under -Fe-P, showing that perturbation of Pi transport to the chloroplast is not associated with the stay-green phenotype. Furthermore, plants lacking key genes that regulate Pi homeostasis, the transcription factor *PHOSPHATE RESPONSE 1 (PHR1)* and the *SPX DOMAIN GENE 1 (SPX1)*, exhibited similar chlorophyll content as wild-type plants in -Fe-P (Supplementary Fig. 9). Given that none of the Pi transport/homeostasis related genes affects chlorophyll accumulation in -Fe-P (Supplementary Fig. 9), *pht4;4* mutant phenotypes indicate that these effects are mediated through its AsA transport activity rather than Pi transport activity. Our conclusion is further supported by the prevention of chlorosis by AsA supplementation in -Fe+P in wild-type plants but not in *pht4;4* mutant plants.

Changes. Main Text: Page 5, Lines 155-164. Supplementary Information: Supplementary Figure 9, Figure Legend: Page 12, Lines 82-91.

3) How did other chlorophyll fluorescence parameters, such as non-photochemical quenching, change under several Fe and Pi conditions?

Response: We have analyzed how Fe and P availability influence non-photochemical quenching (NPQ). Plants under -Fe-P showed slightly lower but stabilized NPQ compared to those in +Fe+P. -Fe+P caused a strong and significant decrease in NPQ observable starting at 76h after the transfer to -Fe+P conditions (Supplementary Fig. 2D).

Changes. Main Text: Page 4, Line 98-100. Supplementary Information: Supplementary Figure 2D, Figure Legend: Page 5, Lines 19-20.

4) Is Fe and Pi deficiency-mediated retrograde signaling associated with genome uncoupled (*gun*) gene?

Response: Our transcriptome analysis did not show any changes in the expression of *GUN1* (AT2G31400) under -Fe+P or -Fe-P conditions. Previous work by Patrice A Salomé, Michele Oliva, Detlef Weigel, and Ute Krämer (EMBO J (2013) 32:511-523)

concluded that: "GUN1 does not play a general role in seedlings grown on various Fe supply". Therefore, it is unlikely that Fe and P deficiency-mediated retrograde signaling is associated with GUN1.

Minor comments:

1) In supplemental Figure 1, the authors mistook typing. [0,--] and [0.--].

Response: Corrected.

Reviewer 3

Review of the manuscript entitled "Interdependent Iron and Phosphorus Availability Controls Photosynthesis Through Retrograde Signaling"

In a previous study, involving some of the authors, the effect of single and multiple nutrient stress due to limited availability of Fe, Pi and Zn was examined. The authors showed that the negative effect of Fe starvation was alleviated by concomitant Pi depletion (or Pi and Zn depletion) in rice and highlighted a possible role of the Pi transporter OsPHO1;1 gene in coordinating pathways involved in Fe transport and Pi (and Zn) signaling (Saenchai et al. 2016. *Plant Sci.* 7:396. doi: 10.3389/fpls.2016.00396). In the present study, the authors focus on the mechanism that underlies the chloroplast response to concomitant Fe and (inorganic) P depletion leading to the 'stay-green' phenotype. Under -Fe-P condition, chlorophyll content was similar to +Fe+P control in different plants, but Fe content and ferritin 1 gene (AtFER1) activity measured on Arabidopsis shoots were low in Fe deficiency both with and without P. Chlorophyll content and PSII function decline were monitored in time course experiments, and three time points better examined by means of RNA sequencing. Then, the study was focused on a sub-set of genes specifically downregulated in -Fe+P and not in -Fe-P, almost all related to chloroplast function, and thus suggesting a relationship to chlorotic phenotype observed only in -Fe+P condition. The analysis of their expression in a worldwide collection of Arabidopsis accessions showed a positively correlation each other. Genome-wide association study (GWAS) was performed to identify 38 QTLs, and in particular the gene PHT4;4, encoding for an ascorbate (ASA) transporter localized at the chloroplast envelop membrane. PHT4;4 has a role in the xanthophyll cycle during photosynthesis and shows enhanced gene expression upon photostress. Experiments with a null pht4;4 allele mutant and the complemented line, and with VTC4 mutant, which reduces the ASA level in leaves, indicated that ASA biosynthesis and transport into the chloroplast is important for the 'stay green' phenotype in Fe deficiency. On the other hand, VTC4 was downregulated in Fe deficiency when P was present, and ASA addition prevented the downregulation of photosynthesis genes. GWAS analysis identified a second gene, encoding for the

transcription factor bZIP58, which was analyzed by qPCR, mutant and complemented lines, GFP-chimera localization.

The photosynthesis genes, as well as ASA biosynthesis mediated by VTC4, appeared to be controlled by bZIP58, which was downregulated in Arabidopsis shoot under -Fe+P. The authors concluded proposing a model where Fe deficiency causes a decrease of ASA content in the chloroplast in the presence of P, with the consequence of a chlorotic phenotype. ROS should accumulate in this condition, with a detrimental effect on bZIP58 itself and, in turn, the photosynthesis genes.

Fe accumulates in the chloroplast playing a pivotal role as cofactor in photosystem complexes and almost all the component of the electron transport chain, despite this the most attention has been paid in the past on Fe deficiency effect at root level. Moreover, plants must adapt to a fluctuating availability of nutrients, and the identification of key genes involved in the co-regulation of micro- and macronutrient homeostasis is of major interest. Thus, the outcomes of this work are interesting, and the conclusions supported.

Moreover, a lot of work using high standard methodologies has been done.

However, the major issue concerning the actual Fe availability in the presence of P and/or the effect of P when Fe is scarce remains to be discussed. Detailed comments are following.

Line 78 and following, Figure 1E, F. It's clear that the focus is on -Fe-P condition, but the increase of Fe content and AtFER1 gene expression under +Fe-P is significantly high, and authors should spend a few words to explain and/or support this finding with citation (metal accumulation in -P is normally observed? Which is the more accredited explanation to this? Which is the phenotype under this condition, as a high Fe content might generate an oxidative stress? etc).

Response: The accumulation of Fe under P deficiency has been consistently observed in several studies (i.e. Misson et al., PNAS 2005; Ward et al., Plant Physiology 2008; Saenchai et al., Front Plant Sci. 2016; Shi et al. Front Plant Sci. 2018). P deficiency causes a decrease in shoot biomass associated with accumulation of anthocyanin (Poirier and Buchner, The Arabidopsis Book: e0024, 2002). A pronounced phenotype is also observed at the root level: inhibition of the primary root elongation, and production of more lateral root and root hairs (Bouain et al., Current Genomics 2018). Remarkably, root growth of plants under simultaneous Fe and P deficiency is similar to root growth of plants grown under +Fe+P (Ward et al., Plant Physiology 2008, Balzergue et al., Nature Comm 2017). Müller and colleagues showed that the inhibition of root growth under -P was caused by high Fe accumulation which generates ROS (Müller et al., Developmental Cell 2015).

A high Fe content under P depletion raises the more general question if Fe availability is higher under P depletion than in P presence, and/or if Fe transport to leaves is higher. P is a macronutrient that could form with Fe scarcely soluble Fe-P compounds. In my opinion, this issue was not correctly addressed in the previous work (Saenchai et al. 2016) and, at least in part, might explain the lack of chlorosis observed also in this work for -Fe-P condition.

Response: We now provide evidence that the stay-green phenotype in -Fe-P is not due to higher accumulation of bioavailable iron in leaves (See response 1 to the Reviewer 1).

Changes: Main Text: Page 3, Lines 73-79; Figure 1, Figure Legend: Page 23, Lines 746-750.

Line 80 and following. If I have understood, plants were grown in Fe-deficient media from the beginning, thus it is not surprising to find similar low Fe content in -Fe+P and -Fe-P. Ferritin are plastid-located proteins, and I'm wondering if it's correct to use FER1 expression as a read-out for 'free Fe in leaves'. As regards the link between Fe total content and FER1 expression, Bournier et al (cited as ref.12) showed that Fe distribution and ferritin gene expression could vary independently from total Fe content in the leaves. Even if AtFER1 expression is lower under -Fe-P compared to -Fe+P condition, this could reflect a lower content of 'free Fe' just in the chloroplast, while Fe content in the outside environment could be higher (in fact, total Fe content in leaves doesn't change in the two conditions). Moreover, we have no information about the actual bioavailability of leaf Fe in the two conditions, and it should be taken into account to explain the 'stay green' phenotype.

Response: We agree with the Reviewer, and we have replaced the *AtFER1* expression data with new data for accumulation of bioavailable Fe in shoots using two methods. Our results showed that the lack of chlorosis under -Fe-P is not associated with more bioavailable Fe in shoots. Please see response 1 to Reviewer 1.

Changes: Main Text: Page 3, Lines 73-79; Figure 1, Figure Legend: Page 23, Lines 746-750.

Line 121. 671 or 673 transcripts? See Venn diagram. 673 and 2434 transcripts: uniquely or specifically differentially regulated?

Response: Thank you. Corrected. 673 transcripts. We used specifically differentially regulated.

Changes: Main Text: Page 4, Line 110. Supplementary Information: Supplementary Figure 4, Figure Legend: Page 7, Lines 34-40.

Line 126 and following. It is not clear to me where the 52 genes come from. Is this a sub-set of downregulated -Fe-P genes among 132 (52h) and 759 (76h) genes, and common to both time points? Same consideration for following sentences.

Response: The 52 downregulated genes in -Fe-P are common to both time points 52h and 76h. The sentence in Line 126 is revised as follows: “The common set of 52 genes that were specifically downregulated by -Fe-P both at 52h and 76h after the transfer (Supplementary Fig. 5A, Supplementary Data) showed an enrichment for ribosomal genes (Supplementary Fig. 5B) while upregulated genes (162 genes, Supplementary Fig. 5C, Supplementary Data) revealed an enrichment for genes involved in cation transport, response to water and ester hydrolysis (Supplementary Fig. 5D, Supplementary Data)”.

Changes: Main Text: Page 4, Lines 114-119. Supplementary Information: Supplementary Figure 5, Figure Legend: Page 8, Lines 43-51.

Line 151 and above. GWAS methodology is not in my area of expertise, and is not clear to me the passage from the subset of 32 genes, the accessions and the identification of 38 QTLs, among which the PHT4;4 containing QTL.

Response: The methodology is as follows:

Step 1. We identified 32 genes that were specifically downregulated by -Fe+P based on our transcriptome data at different time points.

Step 2. We retrieved expression data of the 32 genes in published transcriptome of 727 Arabidopsis accessions.

Step 3. We performed Principal Component Analysis (PCA) to reduce the dimensionality of these expression data.

Step 4. We retained PC1, which explained 89.5% of the variation in expression of these genes across the Arabidopsis accessions.

Step 5. We used the PC1 scores of the accessions to perform a genome-wide association mapping.

Step 6. Our GWA analysis detected 38 QTLs containing 145 candidate genes, based on a 20-kb window per QTL and using a 5% false discovery rate (FDR) threshold.

Step 7. Functional validation of candidate genes of a selected QTL led to the identification of the inorganic phosphate/ascorbate transporter *PHT4;4* (AT4G00370) as a causal gene.

We overhauled the Results section to clarify this.

Changes: Main Text: Page 5, Lines 134-149.

Line 166-203. Results show that transport of ASA into the chloroplast is important in Fe deficiency to recover the ‘stay green’ phenotype also when P is present (Fig. 3B). Fe is a cofactor of the photosystem complexes and almost all the component of the electron transport chain in the thylakoids, it is likely the activation under light of ROS scavenging systems in Fe deficiency. Thus, what is not clear at all is why in -Fe+P

plants the biosynthesis of ascorbate is downregulated and the ascorbate content lower (Figure 3D, E).

Response: This is a very interesting point. We believe, the difference is due to the focus of our study on initial response to Fe deficiency stress in comparison to long-term adaptation. AsA levels have been shown to increase in response to 2 to 3 weeks Fe-deficiency treatments in *Brassica napus* leaves and sugar beet roots (Tewari et al, Environ. Expt. Bot., 2013; Zaharieva and Abadía, Protoplasma, 2003). However, 3 to 7 days Fe-deficiency treatment in Arabidopsis exhibited a slight reduction in AsA accumulation in shoots (Ramírez et al, J. Expt. Bot., 2013). In our study, we measured AsA content 52 hours after Fe deficiency treatment, and we found a significant decrease in AsA accumulation in leaves. It's intriguing to find a consistent reduction of AsA during the early phase of Fe-deficiency. One possibility is that AsA reduction could be one of the earliest signs of Fe deficiency stress. Supporting this hypothesis is a recent finding that an early response to salt stress is repression of AsA biosynthesis by ABI4-mediated repression of VTC2 (Kakan et al, BMC Plant Biology, 2021). Identifying the upstream factors of the signaling pathway we discovered, including AsA biosynthesis, will be an important future direction. We have discussed this point in the Discussion section of the revised manuscript.

Changes: Main Text: Page 10, Lines 322-331.

Line 172. Authors should indicate the gene ID for VTCs genes in the text, or at least in the legend and/or M&M.

Response: The gene IDs for VTCs genes are now indicated in the main text and Figure 3 legend.

Changes: Main Text: Page 6, Lines 168-169. Figure 3, Figure Legend: Page 24, Lines 786-787.

Line 226. The expression value of AT1G13600 in Figure 4A is not consistent with the strong downregulation shown in 4B.

Response: We now present data for 76h in Supplementary Data 2 to be consistent with data shown in 4B. Since, it is difficult to judge the response of PRGs including AT1G13600 to different treatments, we have decided to provide data underlying this experiment as Supplementary Data. As you will see AT1G13600 shows >10-fold repression by -Fe+P, which is roughly equal to what we present in Fig. 4B.

Changes: Supplementary Data 2.

Line 241. ASA addition doesn't rescue the chlorotic phenotype in absence of bZIP58. Besides the control on VTC4, bZIP58 might affect ASA transport into the chloroplast. As a consequence, its downregulation in -Fe+P should result in a low ASA content and the impairment of ROS scavenging system. If authors agree, the model has to be changed indicating a lower ASA level in -Fe -P plants.

Response: We now indicate a lower AsA level in -Fe-P plants in the figure. Furthermore, to provide definitive evidence for the presence of a plastidic signal, ROS, which influences the expression of PRGs and chlorophyll content under -Fe+P by repressing bZIP58, we now expressed bZIP58 under the control of ROS-inducible promoter (*JUNGBRUNNEN1*; *JUB1*) to effectively create an antagonist version of *bZIP58*. The endogenous *bZIP58* in Col-0 is repressed by -Fe+P. However, -Fe+P upregulates *JUB1p::bZIP58* by 2.5-fold in *bzip58* null mutant, demonstrating the sensitivity of the *JUB1* promoter to -Fe+P-induced ROS accumulation (Fig. 5). Notably, a greatly enhanced (4-fold) induction of *JUB1p::bZIP58* is detected in *bzip58pht4;4* double mutant plants in -Fe+P, suggesting a greater chloroplastic ROS signal in the absence of the AsA transporter PHT4;4. Enhanced induction of *JUB1p::bZIP58* by -Fe+P in *bzip58pht4;4* compared to *bzip58* lines indicates that PHT4;4 acts downstream -Fe to influence the expression of the *JUB1p::bZIP58* in the nucleus. Consistently, AsA supplementation in -Fe+P prevents the induction of *JUB1p::bZIP58* in the *bzip58* mutant but not in the *bzip58 pht4;4* double mutant. Again, these results demonstrate that PHT4;4-mediated AsA transport to the chloroplast alters chloroplastic ROS that modulates the expression of *bZIP58* detected by the ROS-inducible promoter system. Finally, we show that under -Fe+P, the expression of *JUB1p::bZIP58* enhances chlorophyll accumulation and expression of PRGs in the *bzip58* mutant. These responses are exacerbated in the *bzip58pht4;4* double mutant expressing *JUB1p::bZIP58* (Fig. 5). These new results demonstrate that changes in chloroplastic ROS in -Fe+P are linked to chlorosis and expression of PRG, which is mediated by a nuclear transcription factor bZIP58.

Changes: Main Text: Figure 6, Page 32; Page 9, Lines 262-281; Figure 5. Figure Legend: Page 25, Lines 822-834.

Line 410. Categories represented in the list are not only biological processes.

Response: Thank you for pointing this out, which is correct. We included enrichments in all three branches of the GO. We have now removed the phrase “biological processes (GO-BP)”.

Changes: Main Text: Figure 2, Figure Legend: Page 24, Line 770.

Line 494 and 685 (M&M), Figure S2A-F. It seems to me that the P value cutoff in the plot and Fold Change value do not correspond to what indicated in the legend and in M&M.

Response: We have now corrected this. The axes are in log scales. Red: $|\log_2\text{FoldChange}| > 1$ and $-\log_{10}P > 6$; Blue: $|\log_2\text{FoldChange}| < 1$ and $-\log_{10}P > 6$; Green: $|\log_2\text{FoldChange}| > 1$ and $-\log_{10}P < 6$; Grey: $|\log_2\text{FoldChange}| < 1$ and $-\log_{10}P < 6$. A default cut-off of $|\log_2\text{FoldChange}| > 1$ and adjusted p-value $< 10^{-6}$ was used.

Changes: Supplementary Information: Supplemental Figure 3, Figure Legend: Page 6, Lines 30-33.

Reviewer #4 (Remarks to the Author):

In this study, authors have discovered that the interdependent effects of Fe and P availability on chlorosis are conserved across monocot and eudicot species, and used RNA-seq experiments and eGWAS to identify two candidate genes (*PHT4;4* and *bZIP58*), which prevent the downregulation of photosynthesis genes leading to the stay-green phenotype under iron-phosphorus deficiency. These findings are very interesting and will help to understand the adaptability of photosynthesis to nutrient utilization. However, there are still a few comments that the authors should address.

Major comments:

1. For the GWAS results, theoretically, any gene within the QTL range may be a potentially functional gene. Authors should explain the basis of screening candidate genes clearly.

Response: In our GWAS analysis, we followed up on the 2 most significant QTLs using a 5% false discovery rate (FDR) threshold. Within these two QTL regions, we examined all the genes in a 20-kb window and stopped when we found phenotypes of *pht4;4* and *bzip58* mutants. Precisely, the first QTL located in chromosome 1 (associated with SNP4653399) contains 9-candidate genes (AT1G13570 to AT1G13610) (Supplementary Fig. 7), and the second QTL located on the chromosome 4 (associated with SNP171674) contains 6 candidate genes (AT4G00355 to AT4G00400) (Supplementary Fig. 7). We systemically tested mutants of all the genes in these QTLs. Out of the 15 candidate genes we examined, knock-out mutants of *bZIP58* and *PHT4;4* exhibited phenotypes under different conditions. We have rewritten this part in Results section to clarify this point.

Change: Main Text: Page 5, Lines 143-149.

2. Although the regulation of *bZIP58* on photosynthesis-related genes was illustrated by phenotypes and transcriptome data of mutants under different nutritional conditions, I suggest that additional molecular experiments, such as protein interaction experiments, are needed to further verify the regulation of photosynthesis-related genes.

Response: To test whether *bZIP58* could “directly” regulate PRGs, we used an established method, quantitative *in planta* transactivation assay, which we routinely employ in our lab to identify target of transcription factors (Bossi et al., BMC Genomics (2017) 18:480). Briefly, selected PRG promoters were fused to a reporter gene (a β -glucuronidase (GUS)) and co-expressed in tobacco leaves with the *bZIP58* TF driven by a 35S promoter. 35S-YFP was used as a negative control. We studied five PRGs (*PHOTOSYSTEM II SUBUNIT T (PSBTN)*; *PHOTOSYSTEM II 5*

kD, *PLASTID TRANSCRIPTIONALLY ACTIVE 16 (PTAC16)*, *LIGHT HARVESTING COMPLEX PHOTOSYSTEM II (LHCB4.1)*, *PHOTOSYSTEM II REACTION CENTER W (PSBW)*) during this analysis. We found that bZIP58 activates the promoter activity of 3 of the 5 genes tested (PSBTN, PHOTOSYSTEM II 5 kD and PTAC16) (Supplementary Fig. 11B). Therefore, bZIP58 can directly activate the expression of some PRGs. These results demonstrate that bZIP58 is a key gene to regulate the expression of PRGs in an Fe-dependent manner. We believe that figuring out how bZIP58 regulates “directly or indirectly” the PRGs is another paper.

Change: Main Text: Page 7, Lines 227-237. Supplementary Information: Supplementary Figure 11, Figure Legend: Page 14, Lines 108-119.

Other comments:

1. Figure 2A, the unit of chlorophyll content on the ordinate is incorrectly marked, the author needs to check carefully.

Corrected.

2. Figure 3A, “-log(P-value)” should be changed to “-log₁₀(P-value)”.

Corrected.

3. Figure 4H, the unit of H₂O₂ content on the ordinate is incorrectly marked, the author needs to check carefully.

Corrected (now is Figure 4G).

REVIEWERS' COMMENTS

Reviewer #1 (Remarks to the Author):

The authors have adequately addressed my concerns.

Reviewer #2 (Remarks to the Author):

Thank you for responding to my suggestions.

The authors showed that iron-phosphorus combined deficiency induces the production and chloroplastic transport of ascorbate, and the ascorbate prevents ROS-induced chlorosis via down-regulation of photosynthesis gene by bZIP58.

I felt that this manuscript is further strengthened by revision.

Further comment:

Response (1): The expression data of the 32 photosynthesis-related genes (PRGs) across different treatments at different time points (39h, 52h, 76h) are presented as a heatmap in Figure 2E. RNA-seq data have been deposited to the NCBI's Gene Expression Omnibus (GEO). Furthermore, we profiled the expression of VTC4 and bZIP58 using qRT-PCR on plants transferred from the control (+Fe+P) condition to: +Fe+P, -Fe+P, or -Fe-P for 39h, 52h, or 76h (Supplementary Fig. 10A). While we find repression of bZIP58 by -Fe+P, no change in the expression of bZIP58 is detected at any of these time points under -Fe-P conditions. VTC4 expression was gradually induced in response to -Fe-P during these time points.

->

In Supplementary Fig. 10A, it is clear that down-regulation of bZIP58 subsequently causes VTC4 up-regulation (ascorbate production). But I can not find whether VTC4 up-regulation is correlated with PHT4;4 expression. Was PHT4;4 expression increased by -Fe-P at 39-76 hr ? This explanation (Our transcriptome data show that -Fe+P did not alter PHT4;4 expression.) will be helpful for reader as shown in Response 2. As PHT4;4 expression was potently induced by strong light, it may be unnecessary to further induce the expression by other stress.

Reviewer #4 (Remarks to the Author):

This revised MS has addressed all my previous concerns. I have no further comments.

REVIEWERS' COMMENTS

Reviewer #2 (Remarks to the Author):

Thank you for responding to my suggestions.

The authors showed that iron-phosphorus combined deficiency induces the production and chloroplastic transport of ascorbate, and the ascorbate prevents ROS-induced chlorosis via down-regulation of photosynthesis gene by bZIP58.

I felt that this manuscript is further strengthened by revision.

Further comment:

Response (1): The expression data of the 32 photosynthesis-related genes (PRGs) across different treatments at different time points (39h, 52h, 76h) are presented as a heatmap in Figure 2E. RNA-seq data have been deposited to the NCBI's Gene Expression Omnibus (GEO). Furthermore, we profiled the expression of VTC4 and bZIP58 using qRT-PCR on plants transferred from the control (+Fe+P) condition to: +Fe+P, -Fe+P, or -Fe-P for 39h, 52h, or 76h (Supplementary Fig. 10A). While we find repression of bZIP58 by -Fe+P, no change in the expression of bZIP58 is detected at any of these time points under -Fe-P conditions. VTC4 expression was gradually induced in response to -Fe-P during these time points.

In Supplementary Fig. 10A, it is clear that down-regulation of bZIP58 subsequently causes VTC4 up-regulation (ascorbate production). But I can not find whether VTC4 up-regulation is correlated with PHT4;4 expression. Was PHT4;4 expression increased by -Fe-P at 39-76 hr ? This explanation (Our transcriptome data show that -Fe+P did not alter PHT4;4 expression.) will be helpful for reader as shown in Response 2. As PHT4;4 expression was potently induced by strong light, it may be unnecessary to further induce the expression by other stress.

Response. We have now provided in Supplementary Figure 10A the expression pattern of *PHT4;4* in response to Fe deficiency and Fe-P deficiency. Our results show that there is no change in *PHT4;4* expression in response to this treatment, confirming our RNAseq data.